# LSA: Layer-wise Sparsity Allocation for Large Language Model Pruning Based on Minimal Linear Reconstruction Error

**Zhiguo Yang**,[*] **Changjian Deng**,[*] **Qinke Chen, Zijing Zhou, Jian Cheng**[†]

School of Information and Communication Engineering, University of Electronic Science and Technology of China

`5733meiruhua@gmail.com`
`cjdeng@std.uestc.edu.cn`
`qinkechen7@gmail.com`
`chouzzing@gmail.com`
`chengjian@uestc.edu.cn`

## Abstract

Deploying large language models (LLMs) on platforms with insufficient computational resources remains a key challenge. Weight pruning is an efficient model compression technique that can reduce model size without retraining LLMs. However, due to the massive number of parameters, it is infeasible to estimate the importance of weights globally, and most prior studies assign a uniform sparsity ratio across all layers. Recent findings reveal that layers contribute unevenly to LLM performance, making it necessary to investigate layer-wise importance. Existing layer-wise sparsity allocation methods, such as OWL and DLP, rely on weight scoring and carefully designed score proxies to estimate layer-wise importance and sparsity ratios, while enforcing identical sparsity to blocks and projection weights within a layer to avoid performance degradation. In this work, we propose layer-wise Sparsity Allocation (LSA) for LLM pruning, which quantifies layer-wise importance by evaluating the minimal linear reconstruction error of each transformer layer under the assumption that 50% of its least important weights are removed. Moreover, our method supports non-uniform sparsity allocation at block- or projection-level granularity within layers, without incurring catastrophic performance degradation. Experimental results demonstrate that LSA maintains high performance at high sparsity levels. At an overall sparsity ratio of 70%, LSA surpasses state-of-the-art methods across language modeling tasks and seven zero-shot tasks. Code is available at https://github.com/BeiYazi0/LSA.

## 1 Introduction

Large Language Models (LLMs) have revolutionized the field of Natural Language Processing (Brown et al., 2020; OpenAI, 2024; Touvron et al., 2023a). Nevertheless, their enormous parameter counts impose severe constraints on deployment due to limited memory and computational resources. Parameter pruning (Hassibi et al., 1993; LeCun et al., 1989; Han et al., 2015) has emerged as a widely used and effective compression technique to alleviate this issue. Recent studies further demonstrate that extensive pruning can be achieved without fine-tuning (Frantar and Alistarh, 2023; Sun et al., 2024a), showing that the parameters of LLMs can be safely removed without retraining.

SparseGPT (Frantar and Alistarh, 2023) and Wanda (Sun et al., 2024a) are two representative one-shot pruning approaches that rely on limited calibration data. They impose uniform sparsity across all layers and enable the model to be deployed without full retraining after pruning, while still achieving competitive inference performance under unstructured sparsity. Nevertheless, these methods overlook the varying importance of different layers within LLMs. Recent works (Kovaleva et al., 2021; Sun et al., 2024b) have shown that LLM activations contain a small number of very

---

[*]Equal contribution
[†]Corresponding author

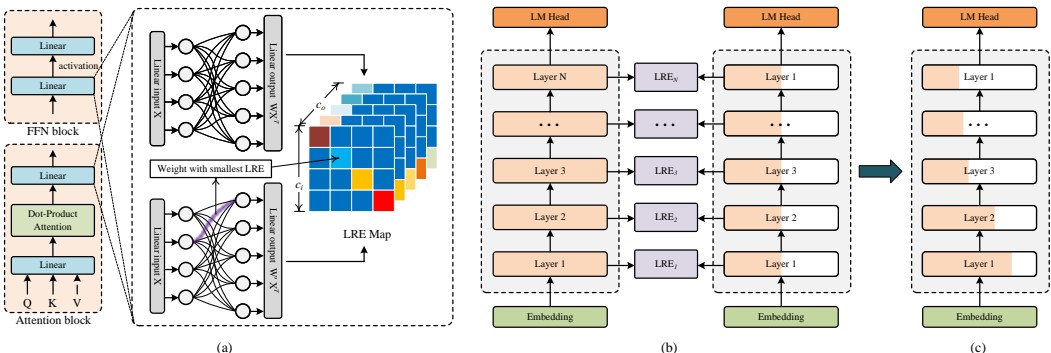

Figure 1: (a) illustrates the weight with minimal linear reconstruction error (LRE) for linear layers within FFN and attention blocks. (b) denotes the layer-wise LRE across all Transformer layers, computed by assuming removing 50% of the weights that contribute least to the reconstruction error in each layer. (c) represents the allocation of different sparsity rates based on the principle that layers with lower reconstruction error should exhibit lower sparsity.

large magnitude features that critically affect model behavior. Dettmers et al. (2022); Puccetti et al. (2022); Lee et al. (2024) indicate that ignoring these outliers leads to severe performance degradation in model compression. Hence, Outlier-Weighted layer-wise Sparsity (OWL) (Yin et al., 2024) allocates non-uniform sparsity ratios across layers according to the layer-wise Outlier Distribution (LOD), which reflects the retention rate of weight outliers. It posits that layers with a higher proportion of weight outliers should be assigned lower sparsity. However, the empirically determined multiplier of the mean pruning score does not generalize well across different model types. In contrast, Dynamic Layer-wise Pruning (DLP) (Chen et al., 2025) demonstrates that using a fixed hyperparameter is suboptimal for LLMs. DLP employs the median of weight scores to characterize layer redundancy and translates this measure into relative layer importance. By leveraging the median, DLP dynamically adapts to both the model's weight distribution and the calibration data, thereby obviating the need for predefined outlier selection criteria.

Both OWL and DLP use Wanda-style weight scoring, computed as the product of weight magnitudes and activation norms, followed by an empirical reduction function to obtain a score proxy for layerwise importance. Although DLP demonstrates that using the median as the reduction function outperforms OWL, which relies on an empirical multiplier on the mean weight score, it has not been established as the optimal reduction function since only a limited set of alternatives has been evaluated. Meanwhile, sparsity allocation at finer granularity than the Transformer layers, such as block level (self-attention and feed-forward blocks) and projection level, has not been sufficiently addressed by either method. We find that when such finer-grained sparsity allocation is applied, the performance of both OWL and DLP degrades substantially.

Hence, we identify two important questions that prior work has underexamined:

- *Is it necessary to identify a superior reduce function for assessing layer importance?*

- *Can non-uniform sparsity be applied at finer granularity without degrading performance?*

To answer these questions, we measure each layer's redundancy by computing the minimal linear reconstruction error between the dense and sparse models. Intuitively, layers with higher errors typically contain more parameters exhibiting high scores (product of weight magnitudes and activation norm) (Yin et al., 2024; Chen et al., 2025) and demonstrate greater tolerance to their removal. This resilience, coupled with their tendency towards uniform error contributions (rather than large outliers) upon weight removal, signifies higher redundancy. Conversely, layers with lower errors, while having a smaller average impact, are more prone to containing weight outliers whose removal causes significant performance drops, reflecting lower redundancy. Accordingly, we transform the redundancy characterized by the layer-wise linear reconstruction error into a relative importance score and allocate lower sparsity ratios to layers with lower redundancy. Furthermore, we extend our method to finer granularity, including block-wise and projection-wise allocation.

In this paper, we propose a novel layer-wise Sparsity Allocation (LSA) method. The LSA pipeline is illustrated in Figure 1. Compared with other sparsity allocation strategies, our LSA avoids the computation of weight scores as well as the empirically designed reduce functions. Our method further enables the assignment of distinct sparsity ratios to different projections within the same Transformer layer, providing a finer-grained allocation that can lead to performance improvements. To the best of our knowledge, this is the first work to explore projection-wise sparsity allocation, providing a novel perspective for future research.

Overall, the contributions of our work are as follows:

- We propose a novel metric for measuring layer-wise importance that evaluates layers directly using linear reconstruction error. This approach removes the need for weight-wise scoring and the manual choice of a reduce function.

- We present an effective sparsity allocation method. Comprehensive experiments show that LSA accurately captures layer importance and consistently outperforms state-of-the-art layerwise sparsity allocation methods for LLMs.

- We demonstrate that LSA applies not only to layer-wise pruning but also to finer-grained sparsity allocations. To the best of our knowledge, LSA is the first approach to achieve finer granularity without catastrophic performance degradation.

## 2 RELATED WORK

### 2.1 LLM PRUNING

LLM pruning can be categorized into unstructured pruning and structured pruning. Unstructured pruning (LeCun et al., 1989; Hassibi et al., 1993; Frantar and Alistarh, 2023; Sun et al., 2024a; Zhang et al., 2024a) removes weights one by one according to a weight-wise importance score. SparseGPT (Frantar and Alistarh, 2023) combines the weight amplitude with the inverse Hessian to form an importance score and performs parameter updates accordingly. Wanda (Sun et al., 2024a) combines the weight amplitude with the L2 norm of input activation as an importance metric and prunes weights that have minimal impact on model performance. In contrast, structured pruning (Ma et al., 2023; Hou et al., 2020; Khaki and Plataniotis; Kurtić et al., 2023; Li et al., 2023; Zhang et al., 2024b) focuses on removing coupled components such as attention heads or neurons to achieve a more regularized, hardware-friendly structure. LLM-Pruner (Ma et al., 2023) estimates weight importance using first-order gradient information and ranks coupled groups within each Transformer layer by their total scores. OSSCAR (Meng et al., 2024) uses the SparseGPT metric to compute channel importance and iteratively removes and updates channels in the pruning list via local search and low-rank inverse Hessian updates. Our work primarily focuses on unstructured pruning.

### 2.2 LAYER-WISE SPARSITY FOR PRUNING

SparseGPT and Wanda both employ uniform pruning, where each layer share the same sparsity level. However, Frankle and Carbin (2018a) shown that different layers contribute unequally to model performance, suggesting that uniform pruning strategies may be suboptimal as they ignore layer-wise importance. Recent studies (Yin et al., 2024; Chen et al., 2025; Gao et al.; Li et al., 2024a; Lu et al., 2024; Huang et al., 2025; Sieberling et al., 2025) have explored non-uniform layer-wise pruning strategies. OWL (Yin et al., 2024) allocates sparsity based on the ratio of weight outliers per layer, linking the outlier distribution to model performance. DLP (Chen et al., 2025) instead uses the median of weight scores as a measure of redundancy. However, both methods rely on the Wanda metric for scoring weights and are limited to layer-wise granularity. In contrast, our method leverages minimal linear reconstruction error to directly estimate layer redundancy, eliminating the need for wight-wise scoring and careful selection of reduce function. Moreover, we explore finer-grained sparsity allocation while maintaining performance comparable to layer-wise.

### 2.3 LINEAR RECONSTRUCTION ERROR IN PRUNING

Our work is also related to methods that minimize linear reconstruction error during pruning (He et al., 2017; Luo et al., 2017; Ding et al., 2019; Zhuang et al., 2018; Meng et al., 2024). Thinet

(Luo et al., 2017) prunes filters by minimizing the output error between selected and full channels. Zhuang et al. (2018) introduces a discrimination-aware loss to jointly reconstruct errors and assess channel importance. However, both approaches require repeated forward passes, often proportional to the square of channel count, making them computationally prohibitive for LLMs. In contrast, our method improves efficiency by precomputing the error caused by removing each channel, greedily pruning the least important one while updating the remaining errors. Moreover, our method is extended to unstructured pruning by partitioning input channels into groups to accelerate pruning.

## 3 PRELIMINARIES

### 3.1 LINEAR RECONSTRUCTION ERROR

Let $\mathbf{W} \in \mathbb{R}^{c_o, c_i}$ be the weight matrix of a fully connected layer, where $c_o$ and $c_i$ denote the number of output and input channels, respectively. Given $N$ calibration samples, the corresponding input activations are $\mathbf{X} \in \mathbb{R}^{N \times L, c_i}$, where $L$ is the sequence length. Pruning is represented by a binary mask $\mathbf{M} \in \{0, 1\}^{c_o, c_i}$ that indicates which weights are removed or retained. The linear reconstruction error after pruning is defined as:

$$\mathbf{E} = \left\| \mathbf{W}\mathbf{X}^T - (\mathbf{M} \odot \mathbf{W})\mathbf{X}^T \right\|_2^2 \tag{1}$$

Pruning for LLMs can be seen as a reconstruction task: given sparsity $p$, remove some pre-trained weights to minimize the output difference between the sparse and dense models. For large models, use hierarchical (layer-wise) reconstruction to split the problem into per-layer subproblems.

### 3.2 WEIGHT OUTLIERS BASED LAYER-WISE SPARSITY ALLOCATION

OWL (Yin et al., 2024) integrates input activation with weight magnitude to identify weight outliers. For the $l$-th layer with input activations $\mathbf{X}^l \in \mathbb{R}^{N \times L, c_i}$, the OWL score is defined as $\mathbf{A}_{i,j}^l = |\mathbf{W}_{i,j}^l| \left\| \mathbf{X}_j^l \right\|_2$. where $|\mathbf{W}_{ij}^l|$ represents the magnitude of the weight, $\left\| \mathbf{X}_j^l \right\|_2$ denotes the $L_2$ norm of the input activations connected to the weight.

OWL introduces the Layer-wise Outlier Distribution (LOD), which quantifies the proportion of weight outliers of layers. An outlier is any weight whose OWL score exceeds the layer mean by a factor of $m$. The LOD is the ratio of these outlier weights to the total number of weights.

$$D^l = \frac{\sum_{i=1}^{c_o} \sum_{j=1}^{c_i} \mathbb{I}(\mathbf{A}_{i,j}^l > m \cdot \overline{\mathbf{A}}^l)}{c_i c_o} \tag{2}$$

where $m$ is a constant, $\overline{\mathbf{A}}^l$ represents the mean of $\mathbf{A}^l$ and $\mathbb{I}(\cdot)$ is the indicator function that equals 1 if its argument is true and 0 otherwise.

After computing the LOD, OWL assigns non-uniform sparsity across layers on the principle that layers with higher LOD are more important and should receive lower sparsity. However, selecting an appropriate $m$ is challenging. DLP (Chen et al., 2025) reports that the optimal $m$ varies across model types and parameter scale. To resolve this, DLP proposes estimating each layer's redundancy and converting those estimates into relative importance scores. Rather than using a fixed threshold, DLP represents the redundancy of the l-th layer by the median of $\mathbf{A}^l$. It normalizes the redundancy scores, subtracts the normalized vector from a vector of ones to obtain importance scores, and then follows OWL's procedure to assign non-uniform sparsity.

## 4 METHODOLOGY

### 4.1 REDUNDANCY REPRESENTATION: MINIMAL LINEAR RECONSTRUCTION ERROR

Let $\mathbf{H} = \mathbf{X}^T\mathbf{X}, \mathbf{S} = \mathbf{H} \odot (\mathbf{W}^T\mathbf{W})$. The set of indices corresponding to the pruned input channels is denoted as $\mathbb{P}$. The linear reconstruction error after structured pruning can be written as (a detailed derivation is provided in Appendix A):

$$\mathbf{E} = \sum_{p=1}^{c_o}\sum_{q=1}^{N\times L}\left(\sum_{i\in\mathbb{P}}\mathbf{W}_{p,i}\mathbf{X}_{q,i}\right)^2 = \sum_{p=1}^{c_o}\sum_{q=1}^{N\times L}\left(\sum_{i\in\mathbb{P}}\sum_{j\in\mathbb{P}}\mathbf{W}_{p,i}\mathbf{W}_{p,j}\mathbf{X}_{q,i}\mathbf{X}_{q,j}\right) = \sum_{i\in\mathbb{P}}\sum_{j\in\mathbb{P}}\mathbf{S}_{i,j} \quad (3)$$

Given a sparsity level $p$, the number of input channels to prune is $c_s = \lfloor c_i \cdot p\rfloor$. To minimize linear reconstruction error, we select a $c_s \times c_s$ submatrix of $\mathbf{S}$ whose entry sum is minimal. However, exhaustive search is not feasible for LLMs, as the total number of such submatrices is given by $C(c_i, c_s)$. Therefore, we initialize a vector $\boldsymbol{\epsilon} \in \mathbb{R}^{c_i}$ with the diagonal entries of $\mathbf{S}$, which represents the error associated with pruning a specific input channel.

By updating $\boldsymbol{\epsilon}$, the linear reconstruction error for pruning input channel $j$, denoted $\epsilon_j$, is calculated as the sum of entries of the submatrix indexed by $\{j\} \cup \mathbb{P}$ minus the sum of entries of the submatrix indexed by $\mathbb{P}$, where $\mathbb{P}$ denotes the set of indices of already pruned input channels. Initially, $\mathbb{P}$ is empty and $\boldsymbol{\epsilon}$ satisfies the constraints. In each step, we greedily select the channel with the smallest $\epsilon_j$. Subsequently, we update $\boldsymbol{\epsilon}$ using the corresponding row and column of the matrix $\mathbf{S}$, and set the selected channel's error to infinity so it cannot be chosen again.

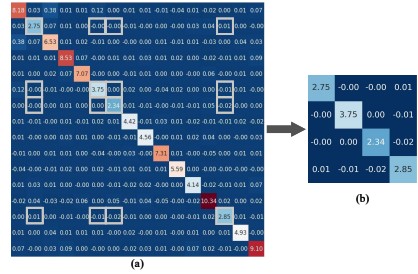

Figure 2: Submatrix selection. (a) $\mathbf{S}$ with 16 input channel. (b) Submatrix indexed by $\{6, 1, 13, 5\}$ has the minimal entry sum at 25% structured sparsity.

Note that $\mathbf{H}$ does not depend on the output channels. Define $\mathbf{S} = \{\mathbf{S}^{(1)}, \mathbf{S}^{(2)}, ...\}$ with $\mathbf{S}^{(k)} = \mathbf{H}\odot(\mathbf{W}_{k,:}^T\mathbf{W}_{k,:})$. By minimizing the entry sum of the submatrices of $\mathbf{S}^{(k)}$, we can perform channel-by-channel pruning, thereby achieving unstructured pruning. The layer linear reconstruction error equals the sum of the per-channel errors, $\sum_{k=1}^{c_o}\sum_{i\in\mathbb{P}^{(k)}}\sum_{j\in\mathbb{P}^{(k)}}\mathbf{S}_{i,j}^{(k)}$, where $\mathbb{P}^{(k)} = \{t|\mathbf{M}_{k,t} = 0\}$. However, computing the full matrix $\mathbf{S}^{(k)}$ for every channel is computationally and memory intensive. A more efficient approach is to compute only the rows of $\mathbf{S}^{(k)}$ that are required for each update, rather than precomputing the entire matrix. Concretely, initialize the matrix $\mathbf{e} \in \mathbb{R}^{c_o\times c_i}$ with $\mathbf{e}_{k,i} = \mathbf{W}_{k,i}^2\mathbf{H}_{i,i}$, then update only the affected entries of $\mathbf{e}_{k,:}$ using the corresponding rows and columns of $\mathbf{H}$ during each pruning step.

$$i = \arg\min_i \mathbf{e}_{k,:}$$
$$\mathbf{e}_{k,:} \leftarrow \mathbf{e}_{k,:} + 2\mathbf{S}_{i,:}^{(k)} = \mathbf{e}_{k,:} + 2\mathbf{W}_{k,i}(\mathbf{W}_{k,:}\odot\mathbf{H}_{i,:}) \quad (4)$$
$$\mathbf{e}_{k,i} \leftarrow \infty$$

To accelerate error computation, we assign the same sparsity level to all output channels. This will be achieved through vectorized parallel pruning in each update step, resulting in an index vector of length $c_o$ that specifies the pruned weight indices for every output channel. To further speed up the computation, we partition the $c_i$ input channels into groups of size $B$ and compute the linear reconstruction error group by group. After processing the current group, we update the corresponding $e$ scores for the remaining unprocessed groups. The pseudo-code to minimize the linear reconstruction error of LSA is shown in Algorithm 1. Detailed implementation can be found in Appendix B.

After applying Algorithm 1, **the model remains dense as we do not prune any weights.** Moreover, we demonstrate that the linear reconstruction error not only allocates layer-wise sparsity but also can provide an importance metric for unstructured pruning. Appendix H compares our resulting metric with Wanda's under a uniform sparsity setting. The results show that pruning by error yields substantially better performance than Wanda. This improvement arises since our metric accounts for the additional error that accumulates when multiple weights are pruned simultaneously, suggesting that using error to compute layer importance is preferable to Wanda-based scores.

## 4.2 Linear Reconstruction Error Based Layer-wise Pruning

For each layer, we measure redundancy using the minimal error calculated by Algorithm 1 with $p = 0.5$. Notably, we find that **the sparsity ratio $p$ used to compute the linear reconstruction error is not a sensitive hyperparameter.** Experiments in Appendix I show that setting $p$ below 70% produces similar performance as the sparsity allocation results are similar. Consequently, we fix $p = 50\%$ in all experiments. Although layers with low linear reconstruction error imply that the average error introduced by reducing individual weights is small, large outliers are more likely to appear. This occurs because a lower value is more susceptible to yield an outlier that is $m$ times greater, indicating low redundancy. Conversely, layers with high linear reconstruction error tend to have a uniformly large error when individual weights are removed, making it difficult to identify additional weight outliers, indicating high redundancy. This conclusion is supported by OWL: layers with high error consistently exhibit high sparsity and low LOD in OWL, as shown in Figure 4.

To properly compare the importance, we normalize the error and convert them into relative importance. The importance score of the l-th layer is defined as $\boldsymbol{I}^l = 1 - \mathbf{E}^l / \sum_i \mathbf{E}^i$.

Intuitively, layers with lower importance scores should receive higher sparsity. However, directly applying raw importance scores often yields poor results since their scale should adapt to different target sparsity ratios $pr$. Inspired by OWL and DLP, we introduce a hyperparameter $\beta$ to rescale importance values according to $pr$. By compressing the importance range to $[0, 2\beta]$, the l-th layer's sparsity $\boldsymbol{s}^l$ is adjusted within $[pr - \beta, pr + \beta]$ by allocating sparsity as $\boldsymbol{s} = pr + \text{mean}(\boldsymbol{d}) - \boldsymbol{d}$, where $\boldsymbol{d} = \boldsymbol{I} \times 2\beta$, thereby limiting excessive pruning and preventing performance collapse.

To fully validate LSA, we investigate how sparsity allocation granularity affects model performance. OWL and DLP adopt layer-wise granularity, assigning a single importance score to all projections within a Transformer layer. In contrast, we introduce two finer granularity: block- and projection-wise. Under block-wise granularity, all projections within a block share the same importance score. Let $\boldsymbol{N}$ denote the number of weights of projections. To maintain the overall sparsity ratio $pr$, we compute the sparsity for each block or projection as $\boldsymbol{s} = (pr \times \boldsymbol{N} + (\text{mean}(\boldsymbol{d}) - \boldsymbol{d}) \times \text{mean}(\boldsymbol{N})) / \boldsymbol{N}$.

---

**Algorithm 1** Pseudocode of Minimizing Linear Reconstruction Error algorithm

**Input:** weight $\mathbf{W}$, Hessian $\mathbf{H} = \mathbf{X}^T \mathbf{X}$, sparsity $p$, number of output channels $c_o$, group size $B$

**Output:** minimal linear reconstruction error $\mathbf{E}$

1: $\mathbf{E} \leftarrow 0$
2: $\mathbf{e} \leftarrow \mathbf{W} \odot \mathbf{W} \times diag(\mathbf{H})$ // initialize linear reconstruction error for each weight
3: $c_s \leftarrow \lfloor B * p \rfloor$
4: **for** $i \leftarrow 0, B, 2B, \ldots$ **do**
5: $\quad \mathbf{Z} \leftarrow \mathbf{0}_{c_o \times B}$
6: $\quad$ **for** $j \leftarrow 0$ to $c_s - 1$ **do**
7: $\qquad \boldsymbol{v} \leftarrow \arg\min \mathbf{e}_{:,i:i+B}$ // get pruned index for each output channel
8: $\qquad \mathbf{E} \leftarrow \mathbf{E} + \sum_{k=1}^{c_o} \mathbf{e}_{k,\boldsymbol{v}_k}$ // accumulate linear reconstruction error
9: $\qquad \mathbf{e}_{:,i:i+B} \leftarrow \mathbf{e}_{:,i:i+B} + 2\mathbf{W}_{:,i:i+B} \times \mathbf{W}_{:,\boldsymbol{v}} \odot \mathbf{H}_{\boldsymbol{v},i:i+B}$ // update for current group
10: $\qquad \mathbf{Z}_{:,\boldsymbol{v}} \leftarrow \mathbf{W}_{:,\boldsymbol{v}}$ // record pruned weight
11: $\quad$ **end for**
12: $\quad \mathbf{e}_{:,i+B:} \leftarrow \mathbf{e}_{:,i+B:} + 2\mathbf{Z}\mathbf{H}_{i:i+B,i+B:}$ // update for dense groups
13: **end for**
14: **return** $\mathbf{E}$

---

## 4.3 Empirical Study

**Evaluation of Finer Granularity.** Although DLP suggests that projection-wise granularity sparsity allocation can cause catastrophic performance degradation by disrupting inter-layer information flow, Table 1 shows that the degradation produced by our projection-wise is less severe than DLP's. For Llama1-7B and Llama2-7B at 70% sparsity with $\beta = 0.15$, both block- and projection-wise allocation reduce performance. However, the performance gap among LSA with different granularity is minimal, whereas OWL and DLP suffer substantially larger declines. As shown in Figure 3, LSA with block-wise granularity is markedly more robust than OWL and DLP. While OWL and DLP are effective only within a narrow $\beta$ range $[0, 0.07]$, LSA remains stable across a much wider interval $[0, 0.17]$, with the resulting sparsity ratios stay within $[0, 1]$. Notably, LSA's block-wise results even surpass layer-wise performance, demonstrating superior tolerance to hyperparameter variation. Moreover, Table 2 shows that LSA with block-wise attains lower perplexity than with

layer-wise at 70% sparsity for LLaMA3. These findings indicate that LSA successfully generalizes to finer-granularity sparsity allocation, whereas both OWL and DLP fail to do so.

**Relationship between Linear Reconstruction Error and Layer Importance.** To further assess the effectiveness of LSA, we measure layer-wise redundancy in LLMs using linear reconstruction error. If errors are balanced across layers, it suggests that our method fail to distinguish layer importance. The bar chart in the background of Figure 4 shows that layers differ in redundancy, indicating unequal contributions to overall model performance. Several recent studies (Gromov et al., 2025a; Sun et al., 2025; Li et al., 2024b) support this observation, reporting that deeper layers are often less critical than expected compared to shallower layers. The overall trend of the sparsity curve across layers indicates that shallow layers are more crucial and therefore require more parameters to preserve the model's core capabilities. In contrast, deeper layers appear to have less impact on overall performance, probably because they process more specialized information (Fan et al., 2024).

Table 1: Comparison of sparsity allocation techniques across three types of granularity on Wiki-Text dataset: layer-, block-, and projection-wise.

| Method | Layer-wise Sparsity | LLaMA1-7B | | | LLaMA2-7B | | |
|---|---|---|---|---|---|---|---|
| | | Layer | Block | Projection | Layer | Block | Projection |
| Dense | - | 5.68 | 5.68 | 5.68 | 5.47 | 5.47 | 5.47 |
| SparseGPT | OWL | 18.98 | 25.93 | 29.87 | 20.68 | 27.39 | 29.91 |
| | DLP | 17.78 | 24.78 | 23.26 | 18.68 | 29.64 | 28.05 |
| | Ours | **17.57** | **18.25** | **19.46** | **18.63** | **20.40** | **21.15** |
| Wanda | OWL | 24.85 | 57.91 | 85.39 | 30.03 | 52.57 | 80.32 |
| | DLP | 20.89 | 40.81 | 52.69 | **22.85** | 59.87 | 117.84 |
| | Ours | **20.66** | **21.60** | **24.82** | 22.89 | **25.55** | **34.56** |

Table 2: Results of block-wise (B) granularity on perplexity using the LLaMA3 model with the WikiText dataset at 70% sparsity. The best performance results are highlighted in bold.

| Method | Layer-wise Sparsity | LLaMA 3-8B | LLaMA 3.2-1B | LLaMA 3.2-3B |
|---|---|---|---|---|
| Dense | - | 6.14 | 9.75 | 7.81 |
| SparseGPT | Uniform | 48.39 | 126.70 | 72.94 |
| | OWL | 37.06 | 164.42 | 57.48 |
| | DLP | 40.12 | 112.29 | 54.86 |
| | Ours | 39.56 | 98.95 | 51.45 |
| | Ours (B) | **32.94** | **87.08** | **46.24** |
| Wanda | Uniform | 114.34 | 437.06 | 149.49 |
| | OWL | 89.72 | 388.79 | 132.15 |
| | DLP | 112.95 | 340.68 | 166.05 |
| | Ours | 98.75 | 306.84 | 154.19 |
| | Ours (B) | **80.25** | **202.01** | **126.01** |

Table 3: Comparison of LOD and Perplexity on the WikiText dataset at 70% sparsity. The best performance result is indicated in bold.

| Model | Method | Layer-wise Sparsity | LOD (%)↑ | PPL↓ |
|---|---|---|---|---|
| LLaMA 1-13B | Dense | - | 19.23 | 5.09 |
| | SparseGPT | Uniform | 178.18 | 20.36 |
| | | OWL | 194.14 | 14.15 |
| | | DLP | 221.73 | 12.87 |
| | | Ours | **233.36** | **12.45** |
| | Wanda | Uniform | 187.71 | 53.47 |
| | | OWL | 203.86 | 16.49 |
| | | DLP | 232.14 | 13.94 |
| | | Ours | **241.14** | **13.65** |
| LLaMA 2-13B | Dense | - | 28.26 | 4.88 |
| | SparseGPT | Uniform | 194.05 | 19.53 |
| | | OWL | 211.46 | 14.31 |
| | | DLP | 237.37 | 13.39 |
| | | Ours | **246.05** | **12.56** |
| | Wanda | Uniform | 199.90 | 45.26 |
| | | OWL | 216.97 | 18.16 |
| | | DLP | 240.00 | 16.80 |
| | | Ours | **248.80** | **15.36** |

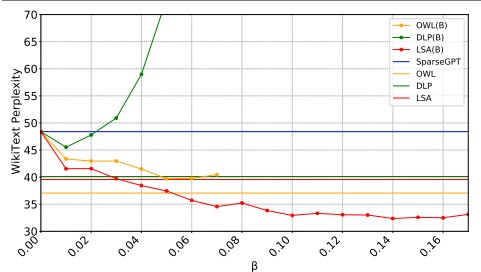

Figure 3: Perplexity of the LLaMA3-8B model on the WikiText dataset, pruned using various $\beta$ at 70% sparsity in block-wise (B).

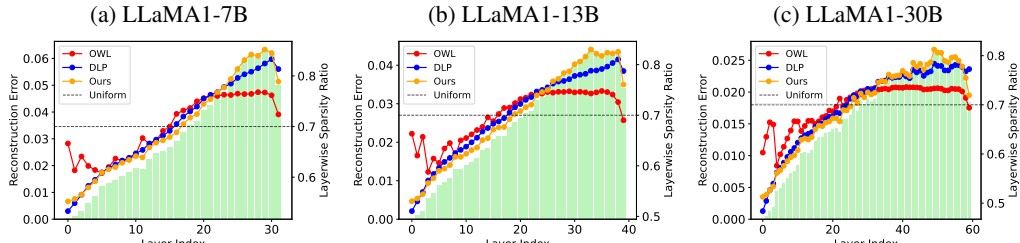

(a) LLaMA1-7B    (b) LLaMA1-13B    (c) LLaMA1-30B

Figure 4: Comparison of layer-wise sparsity distributions. The background bar chart illustrates the normalized linear reconstruction error. In each subplot, the horizontal axis represents the layer index, the left vertical axis denotes the error, and the right vertical axis indicates the layer-wise sparsity.

**Comparison between OWL, DLP and Ours.** To evaluate LSA's ability to preserve weight outliers, we compare the LOD metric of LSA sparsity allocation with Uniform, OWL, and DLP. LOD is

defined as the ratio of weight outliers. For a fair comparison, we use LLaMA1-13B and LLaMA2-13B with $m = 5$ and aggregate LOD in all layers. Table 3 illustrates that all layer-wise methods increase LOD, the proportion of outliers, compared to uniform allocation. Our method preserves outliers most effectively and performs best at high sparsity, achieving the highest LOD and the lowest perplexity. These results demonstrate that our method outperforms both OWL and DLP.

# 5 EXPERIMENTS

## 5.1 EXPERIMENTAL SETUP

**Models and Datasets.** To demonstrate the effectiveness of LSA, we evaluate its performance across various LLMs, including LLaMA1 (7B/13B/30B) (Touvron et al., 2023a), LLaMA2 (7B/13B) (Touvron et al., 2023b), LLaMA3-8B (Grattafiori et al., 2024), as well as LLaMA3.2-1B, LLaMA3.2-3B, Vicuna-7B (Chiang et al., 2023), Mistral-7B (Jiang et al., 2023), and Qwen-7B (Bai et al., 2023). Our evaluation encompasses language modeling proficiency and zero-shot capabilities of sparse LLMs. Language modeling is measured by perplexity on the WikiText (Merity et al., 2017), PTB (Marcus et al., 1994), and C4 (Raffel et al., 2020) validation datasets. For zero-shot evaluation, we report accuracy on seven commonsense benchmarks from EleutherAI LM Harness (Gao et al., 2023), including BoolQ (Clark et al., 2019), PIQA (Bisk et al., 2019), HellaSwag (Zellers et al., 2019), WinoGrande (Sakaguchi et al., 2019), ARC-easy (Boratko et al., 2018), ARC-challenge (Boratko et al., 2018) and OpenbookQA (Mihaylov et al., 2018).

## 5.2 BASELINES.

We select three baseline pruning methods for LLM: Magnitude (JAISWAL et al., 2023), SparseGPT (Frantar and Alistarh, 2023) and Wanda (Sun et al., 2024a). We then assess these baselines when combined with non-uniform sparsity allocation produced by OWL, DLP and LSA. Experiments concentrate on a high sparsity regime, no less than 50%. To ensure statistical reliability, each layer-wise experiment is run at least five times with different seeds, and the average result is reported. For calibration, each method randomly samples 128 examples of 2048 tokens from the C4 dataset.

Table 4: Perplexity results on WikiText dataset with 70% unstructured sparsity across the LLaMA1 and LLaMA2 models. The best performance result is indicated in bold.

Table 5: Comparison of mean zero-shot accuracies (%) for pruned LLaMA1 and LLaMA2 models at 70% unstructured sparsity. The best performance result is indicated in bold.

| Method | Layer-wise Sparsity | LLaMA1 | | | LLaMA2 | |
|---|---|---|---|---|---|---|
| | | **7B** | **13B** | **30B** | **7B** | **13B** |
| Dense | - | 5.68 | 5.09 | 4.10 | 5.47 | 4.88 |
| Magnitude | Uniform | 4.9e4 | 8.5e4 | 9.7e2 | 5.0e4 | 2.1e2 |
| | OWL | 1.9e4 | 2.1e4 | 2.4e2 | 6.0e4 | 59.31 |
| | DLP | **4.1e3** | **7.5e3** | 96.85 | **7.9e3** | 52.16 |
| | Ours | 4.4e3 | 8.7e3 | **93.42** | 9.0e3 | **45.16** |
| SparseGPT | Uniform | 27.18 | 20.36 | 12.34 | 26.32 | 19.53 |
| | OWL | 18.98 | 14.15 | 10.69 | 20.68 | 14.31 |
| | DLP | 17.78 | 12.87 | 9.62 | 18.68 | 13.39 |
| | Ours | **17.57** | **12.45** | **9.49** | **18.63** | **12.56** |
| Wanda | Uniform | 83.45 | 53.47 | 17.67 | 75.01 | 45.26 |
| | OWL | 24.85 | 16.49 | 11.02 | 30.03 | 18.16 |
| | DLP | 20.89 | 13.94 | 9.95 | **22.85** | 16.80 |
| | Ours | **20.66** | **13.65** | **9.77** | 22.88 | **15.36** |

| Method | Layer-wise Sparsity | LLaMA1 | | | LLaMA2 | |
|---|---|---|---|---|---|---|
| | | **7B** | **13B** | **30B** | **7B** | **13B** |
| Dense | - | 58.43 | 61.02 | 62.43 | 59.03 | 60.95 |
| Magnitude | Uniform | 35.51 | 36.25 | 32.82 | 33.66 | 37.20 |
| | OWL | 37.30 | 37.62 | 32.79 | 35.26 | 40.26 |
| | DLP | 38.35 | 38.28 | 40.49 | 37.80 | 44.22 |
| | Ours | **38.66** | **38.65** | **41.00** | **39.17** | **44.38** |
| SparseGPT | Uniform | 42.75 | 45.46 | 52.54 | 41.51 | 45.44 |
| | OWL | 44.23 | 48.16 | 53.53 | 44.49 | 48.18 |
| | DLP | 44.26 | 48.79 | 53.65 | 45.27 | 48.88 |
| | Ours | **44.77** | **48.91** | **53.92** | **45.53** | **49.34** |
| Wanda | Uniform | 37.44 | 39.89 | 50.81 | 35.82 | 38.30 |
| | OWL | 42.95 | 46.98 | 53.42 | 41.11 | 46.94 |
| | DLP | **44.31** | 48.84 | **53.67** | 43.77 | 47.66 |
| | Ours | 44.19 | **48.99** | 53.34 | **43.86** | **47.96** |

## 5.3 MAIN RESULTS

**Language Modelling.** Table 4 reports perplexity of several LLM pruning methods at 70% sparsity on the WikiText dataset. Results for PTB and C4 appear in Appendix K. Additionally, we provide results for different sparsity ratios in Appendix D. Results indicate that LSA is an effective layer-wise pruning method that outperforms both Uniform and OWL across all model sizes. In most cases, LSA achieves lower perplexity than DLP. For example, on LLaMA2-13B, LSA reduces perplexity compared to DLP by 7.00, 0.83, and 1.44 for Magnitude, SparseGPT, and Wanda, respectively.

Table 6: End-to-end decoding latency and throughput of the LLaMA2-7B-chat-hf model using the DeepSparse inference engine with LSA.

| Sparsity | Dense | 10% | 20% | 30% | 40% | 50% | 60% | 70% | 80% | 90% |
|---|---|---|---|---|---|---|---|---|---|---|
| Latency (ms) | 828.76 | 876.48 | 805.76 | 795.49 | 809.78 | 390.84 | 319.61 | 278.24 | 251.17 | 218.23 |
| Throughput (tokens/sec) | 1.21 | 1.14 | 1.24 | 1.26 | 1.23 | 2.56 | 3.13 | 3.59 | 3.98 | 4.58 |
| Speedup | 1.0× | 0.9× | 1.0× | 1.0× | 1.0× | 2.1× | 2.6× | 3.0× | 3.3× | 3.8× |

**Zero-Shot Tasks.** Table 5 reports the average zero-shot accuracy of pruned models at 70% sparsity across seven zero-shot tasks. Additionally, we provide detailed accuracy for each specific task in Appendix L. Overall, LSA achieves the best performance in nearly all settings, demonstrating its promise for more challenging zero-shot downstream tasks. For example, LSA improves average accuracy by 1.37% for Magnitude on LLaMA2-7B and by 0.51% for SparseGPT on LLaMA1-7B.

**Inference Speedup.** To evaluate inference acceleration of sparse LLMs produced by our method, we convert LLaMA2-7B-chat-hf (Touvron et al., 2023b) to ONNX and then measure end-to-end decoding latency using the DeepSparse (Kurtic et al., 2023) inference engine on an Intel (R) Xeon (R) CPU E5-2699 with 44 cores. As shown in Table 6, LSA yields substantial inference speedup compared to dense model. In particular, the speedup exceeds 3.0× at sparsities above 70%.

**Pruning Efficiency.** To evaluate the computational cost of our method, we compare the empirical pruning time of LSA and baselines. We focus exclusively on pruning time, excluding the forward propagation and sparsity allocation, since non-uniform sparsity can be pre-calculated. Table 7 illustrates that the differences between them are minimal. Moreover, as the model scale increases, the pruning time of LSA becomes even lower than that of the uniform baseline. These results indicate that LSA has the potential to accelerate the identification of weights for removal in pruning.

**Performance on Small Models.** We evaluate the performance of LSA and other layer-wise methods on Llama3.2 at 70% sparsity. As shown in Table 2, LSA achieves the lowest perplexity on the Wiki-Text dataset in nearly all cases. Notably, block-wise outperforms layer-wise with LSA, which suggests that **the identical sparsity in layer-wise limits performance improvements derived from sparsity allocation**. Detailed sparsity allocation appear in Appendix F. These results further support the effectiveness of LSA.

Table 7: Comparison of time on LLaMA1 for computing the pruning metric (seconds).

| Method | Layer-wise Sparsity | 7B | 13B | 30B |
|---|---|---|---|---|
| Magnitude | Uniform | 0.56 | 1.16 | 2.54 |
| | Ours | 3.83 | 1.35 | 3.38 |
| SparseGPT | Uniform | 470.39 | 633.85 | 1359.40 |
| | Ours | 411.74 | 622.96 | 1285.27 |
| Wanda | Uniform | 0.83 | 1.52 | 19.44 |
| | Ours | 0.85 | 1.51 | 17.48 |

**Fine-Tuning Performance.** We investigate parameter-efficient fine-tuning (PEFT) techniques to recover the performance of LLMs pruned with LSA. Specifically, we employ a widely recognized PEFT method, LoRA (Hu et al., 2021), and fix pruning mask throughout fine-tuning. Table 8 shows that fine-tuning substantially restores the performance of the pruned models.

Table 8: Perplexity of various LLMs pruned by LSA using Wanda on the Wikitext datasets.

| Model | Method | Sparsity | Perplexity |
|---|---|---|---|
| LLaMA1-7B | Without FT | 0.7 | 20.66 |
| LLaMA1-7B | With FT | 0.7 | 12.27 |
| LLaMA2-7B | Without FT | 0.7 | 22.89 |
| LLaMA2-7B | With FT | 0.7 | 11.75 |

**Performance on More Advanced LLMs.** We evaluate LSA on additional advanced LLMs. As shown in Table 9, LSA reduces perplexity at multiple sparsity levels and consistently outperforms uniform sparsity. Table 10 reports zero-shot accuracy for three models. On Qwen3-8B, OWL and DLP suffer substantially greater degradation than uniform sparsity when pruned with SparseGPT, whereas LSA exhibits only a minor drop. After applying a projection-wise variant of LSA, performance improves further and exceeds that of uniform sparsity, indicating that our method better captures redundancy than OWL and DLP. A similar pattern appears for Qwen2.5-7B, we also adopt a block-wise allocation scheme to address the limitations of layer-wise granularity. Overall, these results demonstrate LSA's ability to generalize across LLM architectures.

Table 9: WikiText perplexity of various LLMs pruned by Uniform and Ours using Wanda.

| Model | Method | 60% | 70% | 80% |
|---|---|---|---|---|
| Vicuna-7B | Uniform | 13.12 | 59.78 | 2195.60 |
| | Ours | 11.63 | 27.63 | 365.03 |
| Mistral-7B | Uniform | 11.22 | 60.89 | 337.36 |
| | Ours | 9.96 | 31.82 | 224.99 |
| Qwen-7B | Uniform | 28.71 | 586.11 | 208755.20 |
| | Ours | 21.27 | 190.78 | 8799.00 |

**Integration with Other Compression Methods.** Although our method focused on unstructured pruning in previous sections, it can also be applied to hardware-friendly settings. In Appendix C we combine it with structured pruning, N:M sparsity, and quantization.

**Integration with More Advanced Pruning Methods** We also evaluate LSA on advanced LLM pruning methods. ADMM-Grad (Boža, 2024) proposes using the Alternating Direction Method of Multipliers (ADMM) for pruning, which is more effective than prior state-of-the-art methods such as SparseGPT and Wanda. Therefore, we combine ADMM-Grad with non-uniform sparsity to further validate LSA. As shown in Table 11, LSA maintains superior performance relative to all baselines.

## 6 CONCLUSION

In this paper, we introduce a linear reconstruction error based layer-wise pruning method that directly measures layer importance without requiring wight-wise scoring or reduce function design. We compute the minimal reconstruction error at 50% sparsity to assess redundancy and convert these values into relative layer importance. Layers with lower importance are assigned higher sparsity. Notably, our method achieves substantial performance gains, outperforming existing state-of-the-art approaches. Moreover, our method successfully generalizes to finer granularity sparsity allocation, maintaining competitive performance compared to layer-wise approaches. These results highlight the promise of finer granularity sparsity allocation and suggest several directions for future work.

Table 10: Accuracy (%) of more advanced models on seven zero-shot tasks at 70% unstructured sparsity with various sparsity allocation methods, including block-wise (B) and projection-wise(C) granularity. The best performance result is indicated in bold.

| Model | Method | Layer-wise Sparsity | WinoGrande | HellaSwag | BoolQ | PIQA | OBQA | ARC-e | ARC-c | Mean |
|---|---|---|---|---|---|---|---|---|---|---|
| LLaMA3-8B | Dense | - | 72.61 | 60.17 | 81.59 | 79.65 | 34.8 | 80.09 | 50.17 | 65.58 |
| | SparseGPT | Uniform | 57.30 | 33.74 | 66.39 | 62.89 | 15.00 | 44.95 | 22.01 | 43.18 |
| | | OWL | 60.93 | 36.67 | 70.58 | 65.40 | 16.20 | 48.27 | 23.81 | 45.98 |
| | | DLP | 61.56 | 37.54 | 70.67 | 65.13 | **19.20** | 48.19 | **25.60** | 46.84 |
| | | Ours | **62.12** | **37.93** | **74.53** | **65.89** | 18.60 | **48.95** | 25.09 | **47.59** |
| | Wanda | Uniform | 48.22 | 27.28 | 50.43 | 55.60 | 13.60 | 32.15 | 17.66 | 34.99 |
| | | OWL | 49.49 | 28.40 | 61.50 | 57.83 | 13.40 | 35.52 | 17.66 | 37.69 |
| | | DLP | 52.41 | 29.51 | 58.53 | 60.66 | 14.00 | 38.51 | **19.03** | 38.95 |
| | | Ours | **53.91** | **30.45** | **60.49** | **60.83** | **15.00** | **41.04** | 19.03 | **40.11** |
| Qwen2.5-7B | Dense | - | 73.01 | 60.04 | 85.11 | 78.78 | 33.20 | 80.47 | 47.78 | 65.48 |
| | SparseGPT | Uniform | 61.72 | 40.00 | 73.24 | 68.93 | 20.00 | 63.05 | 29.18 | 50.88 |
| | | OWL | 61.09 | 38.02 | 64.62 | 67.63 | 19.20 | 59.93 | 27.39 | 48.27 |
| | | DLP | 61.96 | 38.31 | 67.80 | 65.40 | 18.80 | 55.98 | 26.28 | 47.79 |
| | | Ours | 62.67 | 39.04 | **77.40** | 65.78 | 19.00 | 56.10 | 27.65 | 49.66 |
| | | Ours(B) | **64.33** | **40.68** | 72.69 | **68.99** | **23.00** | **63.34** | **29.69** | **51.82** |
| | Wanda | Uniform | 53.04 | 30.59 | 62.02 | 61.81 | 15.80 | 45.75 | 20.56 | 41.37 |
| | | OWL | 52.17 | 30.68 | 62.02 | 62.02 | 14.60 | 45.66 | 19.37 | 40.95 |
| | | DLP | 56.67 | 33.38 | 62.23 | 63.00 | 16.60 | 45.62 | 21.16 | 42.67 |
| | | Ours | **56.91** | **33.41** | **62.57** | **63.06** | **16.80** | **46.68** | **23.29** | **43.24** |
| Qwen3-8B | Dense | - | 67.64 | 57.12 | 86.64 | 76.88 | 31.00 | 83.54 | 55.89 | 65.53 |
| | SparseGPT | Uniform | 62.12 | 38.57 | **73.30** | 68.72 | 21.20 | 61.45 | 29.52 | 50.70 |
| | | OWL | 60.22 | 37.17 | 66.54 | 67.41 | 21.40 | 58.75 | 27.73 | 48.46 |
| | | DLP | 63.77 | 37.58 | 66.00 | 67.49 | 20.40 | 57.49 | 29.61 | 48.31 |
| | | Ours | **64.72** | 38.31 | 68.44 | 67.74 | 22.40 | 59.93 | 31.31 | 50.41 |
| | | Ours(B) | 64.25 | 39.13 | 70.98 | 68.44 | **24.00** | 61.57 | 30.80 | 51.31 |
| | | Ours(C) | 64.01 | **39.49** | 71.96 | **69.15** | 23.80 | **62.92** | **31.48** | **51.83** |
| | Wanda | Uniform | 53.51 | 30.53 | 62.32 | 61.15 | 15.00 | 50.04 | 21.25 | 41.97 |
| | | OWL | 52.09 | 29.52 | 61.99 | 61.15 | 15.20 | 47.18 | 18.86 | 40.86 |
| | | DLP | 55.49 | 31.90 | 62.20 | 62.02 | 16.20 | 48.48 | 23.38 | 42.81 |
| | | Ours | **57.54** | **32.84** | **62.39** | **63.76** | **16.40** | **51.60** | **24.15** | **44.10** |

Table 11: Perplexity on different validation datasets and accuracy (%) on seven zero-shot tasks of LLaMA3-8B models at 70% sparsity using ADMM-Grad pruning with various layer-wise sparsity methods. The best performance result is indicated in bold.

| Layer-wise Sparsity | Perplexity↓ | | | Accuracy(%)↑ | | | | | | | |
|---|---|---|---|---|---|---|---|---|---|---|---|
| | WikiText | PTB | C4 | WinoGrande | HellaSwag | BoolQ | PIQA | OBQA | ARC-e | ARC-c | Mean |
| Uniform | 29.20 | 48.82 | 37.95 | 59.98 | 35.63 | 68.56 | 64.36 | 18.00 | 48.78 | 21.93 | 45.32 |
| OWL | **27.36** | **41.77** | 32.34 | 62.83 | 38.06 | 72.81 | 65.07 | 18.20 | **52.86** | 24.91 | 47.82 |
| DLP | 28.26 | 45.77 | 31.12 | 63.30 | 38.81 | 72.84 | 66.10 | 20.00 | 51.14 | 24.49 | 48.10 |
| Ours | 27.97 | 45.62 | **30.09** | **64.80** | **39.29** | **74.46** | **66.38** | **21.60** | 51.05 | **25.77** | **49.05** |

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

## A  MINIMIZE LINEAR RECONSTRUCTION ERROR FOR STRUCTURED PRUNING

**Efficient Representation of Linear Reconstruction Error**. In structured pruning, each row or column of the mask matrix $\mathbf{M}$ is entirely composed of either ones or zeros. Denote the $i$-th column of $\mathbf{W}$ by $\mathbf{W}_{:,i}$, the $i$-th column of $\mathbf{X}$ by $\mathbf{X}_{:,i}$, and the $i$-th column of $\mathbf{M}$ by $\mathbf{M}_{:,i}$. By setting all elements of $\mathbf{M}_{:,i}$ as 0, we can prune $\mathbf{W}_{:,i}$. The corresponding linear reconstruction error can be expressed as:

$$
\begin{aligned}
\mathbf{E} &= \left\| (\mathbf{W} - [\mathbf{W}_{:,1}, ..., \mathbf{W}_{:,i-1}, \mathbf{0}, \mathbf{W}_{:,i+1}..., \mathbf{W}_{:,c_i}]) \mathbf{X}^T \right\|_2^2 \\
&= \sum_{p=1}^{c_o} \sum_{q=1}^{N \times L} (\mathbf{W}_{:,i} \mathbf{X}_{:,i}^T)_{p,q}^2 \\
&= \sum_{p=1}^{c_o} \sum_{q=1}^{N \times L} \mathbf{W}_{p,i}^2 \mathbf{X}_{q,i}^2 \\
&= \sum_{p=1}^{c_o} \mathbf{W}_{p,i}^2 \sum_{q=1}^{N \times L} \mathbf{X}_{q,i}^2 \\
&= \mathbf{X}_{:,i}^T \mathbf{X}_{:,i} \mathbf{W}_{:,i}^T \mathbf{W}_{:,i}
\end{aligned}
\tag{5}
$$

By setting both $\mathbf{M}_{:,i}$ and $\mathbf{M}_{:,j}$ to 0, we prune the columns $\mathbf{W}_{:,i}$ and $\mathbf{W}_{:,j}$. The corresponding linear reconstruction error can be expressed as:

$$
\begin{aligned}
\mathbf{E} &= \sum_{p=1}^{c_o} \sum_{q=1}^{N \times L} (\mathbf{W}_{:,i} \mathbf{X}_{:,i}^T + \mathbf{W}_{:,j} \mathbf{X}_{:,j}^T)_{p,q}^2 \\
&= \sum_{p=1}^{c_o} \sum_{q=1}^{N \times L} (\mathbf{W}_{p,i} \mathbf{X}_{q,i} + \mathbf{W}_{pj} \mathbf{X}_{qj})^2 \\
&= \sum_{p=1}^{c_o} \sum_{q=1}^{N \times L} (\mathbf{W}_{p,i}^2 \mathbf{X}_{q,i}^2 + \mathbf{W}_{pj}^2 \mathbf{X}_{qj}^2 + 2\mathbf{W}_{p,i} \mathbf{W}_{pj} \mathbf{X}_{q,i} \mathbf{X}_{qj}) \\
&= \mathbf{X}_{:,i}^T \mathbf{X}_{:,i} \mathbf{W}_{:,i}^T \mathbf{W}_{:,i} + \mathbf{X}_{:,j}^T \mathbf{X}_{:,j} \mathbf{W}_{:,j}^T \mathbf{W}_{:,j} + \mathbf{X}_{:,i}^T \mathbf{X}_{:,j} \mathbf{W}_{:,i}^T \mathbf{W}_{:,j} + \mathbf{X}_{:,j}^T \mathbf{X}_{:,i} \mathbf{W}_{:,j}^T \mathbf{W}_{:,i}
\end{aligned}
\tag{6}
$$

Let $\mathbf{S}_{\mathbf{W}} = \mathbf{W}^T \mathbf{W}$, $\mathbf{H} = \mathbf{X}^T \mathbf{X}$, and $\mathbf{S} = \mathbf{H} \odot \mathbf{S}_{\mathbf{W}}$, then the above expression can be rewritten as follows:

$$
\mathbf{E} = \mathbf{H}_{ii} \mathbf{S}_{\mathbf{W}ii} + \mathbf{H}_{jj} \mathbf{S}_{\mathbf{W}jj} + \mathbf{H}_{ij} \mathbf{S}_{\mathbf{W}ij} + \mathbf{H}_{ji} \mathbf{S}_{\mathbf{W}ji} = \mathbf{S}_{ii} + \mathbf{S}_{jj} + \mathbf{S}_{ij} + \mathbf{S}_{ji}
\tag{7}
$$

Let $\mathbb{P}$ denote the set of indices corresponding to the pruned input channels. Based on Eq. (6) and Eq. (7), it can be demonstrated that the linear reconstruction error can be represented as:

$$\begin{aligned}
\mathbf{E} &= \sum_{p=1}^{c_o} \sum_{q=1}^{N \times L} \left( \sum_{i \in \mathbb{P}} \mathbf{W}_{:,i} \mathbf{X}_{:,i}^T \right)_{p,q}^2 \\
&= \sum_{p=1}^{c_o} \sum_{q=1}^{N \times L} \left( \sum_{i \in \mathbb{P}} \mathbf{W}_{p,i} \mathbf{X}_{q,i} \right)^2 \\
&= \sum_{p=1}^{c_o} \sum_{q=1}^{N \times L} \left( \sum_{i \in \mathbb{P}} \sum_{j \in \mathbb{P}} \mathbf{W}_{p,i} \mathbf{W}_{pj} \mathbf{X}_{q,i} \mathbf{X}_{qj} \right) \\
&= \sum_{i \in \mathbb{P}} \sum_{j \in \mathbb{P}} \mathbf{S}_{ij}
\end{aligned} \tag{8}$$

We call this an efficient representation of the linear reconstruction error since it enables computing the error for any index set $\mathbb{P}$ without requiring multiple forward passes.

**Greedy Strategy**. Given a target sparsity level $p$, the number of input channels to be pruned is defined as $c_s = \lfloor c_i * p \rfloor$. To minimize the linear reconstruction error, we select a $c_s \times c_s$ submatrix of $\mathbf{S}$ whose entries have the smallest total sum. In principle, recursion can enumerate all possible submatrices to identify the one with the minimal entry sum, and the corresponding indices provide the optimal pruning set. However, this approach is impractical for large models, since the total number of candidate submatrices is given by $C(c_i, c_s)$. For instance, for the smallest 7B model of LLaMA2 with hidden_size $= 4096$, pruning just four channels would require examining 11,710,951,848,960 submatrices.

One feasible strategy is to utilize the diagonal elements of matrix $\mathbf{S}$ as an importance measure vector $\epsilon \in \mathbb{R}^{c_i}$, which reflects the linear reconstruction error when pruning a specific input channel. After removing channels in $\mathbb{P}$, updating $\epsilon$ makes $\epsilon_j$ equal to the sum of entries of the submatrix indexed by $\{j\} \cup \mathbb{P}$ minus the sum of entries of the submatrix indexed by $\mathbb{P}$.

Initially, the set $\mathbb{P}$ is empty and the importance vector $\epsilon$ satisfies the prescribed constraints. At each pruning step, we greedily remove the channel with the smallest linear reconstruction error score, and update $\epsilon$ by adding the corresponding row and column of $\mathbf{S}$. We then mark the linear reconstruction error of the selected channel as infinite (i.e., adding this channel to the set $\mathbb{P}$), thus preventing its re-selection during subsequent pruning.

$$i = \arg\min_i \epsilon \tag{9}$$

$$\epsilon \leftarrow \epsilon + \mathbf{S}_{i,:} + \mathbf{S}_{:,i} = \epsilon + 2\mathbf{S}_{i,:} \tag{10}$$

$$\epsilon_i \leftarrow \infty \tag{11}$$

Eq. (10) utilizes the symmetric property of $\mathbf{S}$. It is straightforward to verify that employing Eq. (9), (10), and (11) to select channels for inclusion in set $\mathbb{P}$ and to update $\epsilon$ ensures that, whenever channel $j$ is pruned, the linear reconstruction error $\epsilon_j$ equals to the sum of entries of the submatrix indexed by $\{j\} \cup \mathbb{P}$ minus the sum of entries of the submatrix indexed by $\mathbb{P}$.

# B  PYTORCH CODE IMPLEMENTATION: CALCULATE MINIMAL LINEAR RECONSTRUCTION ERROR

```python
def fix_prune(weight, inputs, p=0.5, block_size=128):
    co, ci = weight.shape
    inputs = inputs.reshape((-1, ci))
    sx = inputs.T @ inputs

    ## initialize linear reconstruction error for each weight
    score = (weight ** 2) * torch.diag(sx)
```

```python
re_construct = 0.
prune_num = int(ci * p)
prune_idx = []
for i1 in range(0, ci, block_size):
  i2 = min(i1 + block_size, ci)

  w1 = weight[:, i1:i2] # weight in current group
  w2 = weight[:, i2:] # weight in remain groups
  score1 = score[:, i1:i2] # error in urrent group
  score2 = score[:, i2:] # error in remain groups
  sx1 = sx[i1:i2, i1:i2] # update hessian for current group
  sx2 = sx[i1:i2, i2:] # update hessian for remain groups

  err = torch.zeros_like(w1)
  for i in range(prune_num):
    ## get index of weight to be pruned for each output channel
    idx = torch.argmin(score1, dim=1).unsqueeze(1)
    ## accumulate linear reconstruction error
    re_construct += torch.sum(score1.gather (1, idx))
    prune_idx.append(idx + i1)

    w = w1.gather(1, idx) # gather selected weights
    change = w1 * w * sx1[idx.squeeze(1)]
    score1 += 2 * change # update
    score1.scatter_(dim=1, index=idx, value=torch.inf) # mark

    ## record weight for updating error in remain groups
    err.scatter_(dim=1, index=idx, src=w)

  change = w2 * (err @ sx2)
  score2 += 2 * change # update

return re_construct
```

## C INTEGRATION WITH OTHER COMPRESSION METHODS

Experiments in the previous sections demonstrated LSA's ability to distinguish layer importance in unstructured pruning. To unleash the potential of LSA in more scenarios, we explore the application of LSA in structured pruning, N:M sparsity and quantization by combining LSA with these compression methods.

### C.1 INTEGRATION WITH STRUCTURED PRUNING

Although LSA is primarily designed for unstructured pruning, it extends naturally to structured pruning. Following the approach of LLM-Pruner (Ma et al., 2023), which removes coupled components such as attention heads or neurons to produce a more regular structure and enable acceleration, we use BookCorpus (Zhu et al., 2015) as the calibration dataset and assess the performance of LLaMA1-7B combined with our method. As shown in Table 12, LSA successfully improves performance of structured pruning by allocating non-uniform sparsity across layers.

### C.2 INTEGRATION WITH N:M SPARSITY

We also apply LSA to N:M sparsity. Following the approach presented in DominoSearch (Sun et al., 2021), we adopt a mixed N:8 configuration: rather than using the same N for every layer, we assign different N values across layers while preserving the overall sparsity. Since layer-wise sparsity differences among allocation methods are statistically insignificant and the integer quantization effect of N causes all methods to yield near-identical results, we omit OWL and DLP from the reported results. Table 13 shows that the mixed N:M configuration combined with LSA consistently outperforms uniform N:M allocation. Notably, in high sparsity regimes with 3:8 and 2:8 configurations, LSA yields substantial gains, approximately 2× and 15× improvements in performance, respectively.

Table 12: Perplexity of the LLaMA1-7B models on WikiText and PTB dataset using LLM-Pruner. Best performance indicated in bold.

| Dataset | Layer-wise Sparsity | 20% | 40% | 60% |
|---------|---------------------|-----|-----|-----|
| WikiText | Uniform | 11.04 | 269.36 | 1397.30 |
| | OWL | 9.96 | **152.80** | 3792.64 |
| | DLP | 10.73 | 166.49 | 1043.75 |
| | Ours | **9.23** | 157.92 | **371.97** |
| PTB | Uniform | 256.94 | 1036.08 | 3549.23 |
| | OWL | 139.67 | 1418.18 | 4011.72 |
| | DLP | **65.67** | 914.04 | 3093.68 |
| | Ours | 161.69 | **752.84** | **1240.52** |

Table 13: Perplexity of mixed N:M sparsity (where N refers to non-zero weights) with LLaMA1-7B on the WikiText dataset. The best performance result is indicated in bold.

| Layer-wise Sparsity | N:M Sparsity Structure | | |
|---------------------|------|-------|------|
| | 2:8 | 3:8 | 4:8 |
| Uniform | 7397.19 | 43.68 | 8.65 |
| Ours | **490.41** | **23.60** | **8.39** |

## C.3 INTEGRATION WITH QUANTIZATION

We further compare quantized models pruned with uniform and nonuniform sparsity using LSA. Specifically, We apply the advanced quantization approach, GPTQ (Frantar et al., 2023), to LLaMA1-7B at 70% sparsity level. We evaluate perplexity on Wikitext, PTB, and C4 datasets with bit widths of 2, 4, and 8. The results are presented in Table 14. After quantization, models pruned with LSA outperform those with uniform sparsity at all bit widths. Notably, on WikiText the perplexity using LSA at 4 bits increases by only $1.04\times$ relative to 8 bits, whereas uniform sparsity yields a $3.28\times$ increase.

Table 14: Perplexity of LLaMA1-7B on different validation datasets under varying quantization levels at 70% unstructured sparsity, pruned with LSA using SparseGPT. The best performance result is indicated in bold.

| Bits | Layer-wise Sparsity | Sparsity | WikiText | PTB | C4 |
|------|---------------------|----------|----------|-----|-----|
| 8 | Dense | 0 | 6.18 | 45.81 | 8.25 |
| 8 | Uniform | 70% | 34.35 | 416.76 | 32.53 |
| 8 | Ours | 70% | **24.22** | **370.32** | **24.52** |
| 4 | Dense | 0 | 6.64 | 48.24 | 8.85 |
| 4 | Uniform | 70% | 112.65 | 820.77 | 94.71 |
| 4 | Ours | 70% | **25.25** | **274.01** | **28.26** |
| 3 | Dense | 0 | 8.42 | 109.75 | 11.33 |
| 3 | Uniform | 70% | 142.67 | 1246.90 | 112.66 |
| 3 | Ours | 70% | **72.60** | **639.25** | **64.63** |

## D PERFORMANCE AT VARIOUS SPARSITY RATIOS

To further evaluate the effectiveness of our method, we conducted extensive experiments on the WikiText dataset at various levels of sparsity, from low sparsity ($< 50\%$) to very high sparsity (80%). We compare our method with other layer-wise sparsity techniques, including Global (Frankle and Carbin, 2018b), ER (Mocanu et al., 2018), ER-Plus (Liu et al., 2022), AlphaPruning(Lu et al., 2024), OWL and DLP. As shown in Table 15, all methods perform similarly at sparsity levels below 40%, with only modest differences in perplexity. Notably, LSA consistently outperforms OWL and DLP

at sparsity levels of 70% and above. These results strongly support the effectiveness and applicability of our method.

Table 15: Perplexity of LLaMA1-7B on the WikiText validation dataset with various layer-wise sparsity using Wanda. The best performance result is indicated in bold.

| Method | Sparsity (Dense 5.68) | | | | | | | |
|--------|------|------|------|------|------|----------|----------|-----------|
| | 10% | 20% | 30% | 40% | 50% | 60% | 70% | 80% |
| Global | 5.82 | 6.11 | 7.02 | 9.83 | 49.07 | 35477.43 | 28941.62 | 79951.92 |
| Uniform | 5.70 | 5.82 | **5.99** | 6.40 | 7.26 | 10.89 | 83.45 | 6548.71 |
| ER | 5.70 | 5.81 | 6.04 | 6.59 | 7.82 | 12.47 | 120.50 | 7831.16 |
| ER-Plus | 5.71 | 5.86 | 6.26 | 7.27 | 12.21 | 766.84 | 18190.85 | 185918.25 |
| OWL | 5.71 | 5.81 | 6.02 | 6.39 | 7.22 | 9.44 | 24.85 | 1024.15 |
| AlphaPruning | **5.69** | 5.81 | 6.01 | 6.39 | 7.25 | 9.51 | 24.02 | 775.43 |
| DLP | 5.70 | 5.81 | 6.00 | **6.38** | **7.19** | 9.45 | 20.89 | 552.53 |
| Ours | 5.70 | **5.81** | 6.01 | 6.39 | 7.20 | **9.38** | **20.66** | **484.93** |

The returns on performance improvements diminish with decreasing sparsity, which aligns with common patterns observed in model pruning and compression. Consequently, we focused our experiments on 70% sparsity to highlight performance differences between methods. Still, We further provide zero-shot performance results for Qwen2.5-7B at 60% sparsity in Table 16. These results demonstrate that LSA still achieves noticeable improvements even at moderate sparsity levels.

Table 16: Accuracy (%) of more advanced models on seven zero-shot tasks at 60% unstructured sparsity with various sparsity allocation methods, including block-wise (B). The best performance result is indicated in bold.

| Model | Method | Layer-wise Sparsity | WinoGrande | HellaSwag | BoolQ | PIQA | OBQA | ARC-e | ARC-c | Mean |
|-------|--------|---------|------------|-----------|-------|------|------|-------|-------|------|
| | Dense | - | 73.01 | 60.04 | 85.11 | 78.78 | 33.20 | 80.47 | 47.78 | 65.48 |
| Qwen2.5-7B | SparseGPT | Uniform | 68.03 | 49.47 | 82.14 | **74.65** | 26.00 | 73.57 | 39.76 | 59.09 |
| | | OWL | 68.51 | 48.55 | 81.77 | 74.10 | 26.80 | 73.11 | 38.14 | 58.71 |
| | | DLP | 68.67 | 48.24 | 81.65 | 72.63 | 26.20 | 71.76 | 37.71 | 58.12 |
| | | Ours | 67.25 | 48.57 | 81.96 | 73.07 | 26.40 | 72.26 | 39.93 | 58.49 |
| | | Ours(B) | **69.93** | **49.88** | **83.03** | 73.50 | **28.00** | **73.82** | **40.53** | **59.81** |
| | Wanda | Uniform | 66.54 | 45.67 | 79.91 | 72.03 | 25.20 | **71.84** | 36.60 | 56.83 |
| | | OWL | 67.40 | 44.53 | 78.35 | 71.55 | 24.20 | 71.34 | 36.35 | 56.25 |
| | | DLP | 66.77 | 44.88 | 76.73 | 70.95 | 25.20 | 69.95 | 36.60 | 55.87 |
| | | Ours | 66.61 | 45.38 | 79.24 | 71.49 | 25.20 | 70.03 | 37.54 | 56.50 |
| | | Ours(B) | **69.06** | **46.35** | **81.71** | **72.52** | **25.80** | 71.72 | **38.74** | **57.99** |

## E  COMPARISON WITH MORE LAYER-WISE SPARSITY METHODS

To further validate LSA, we compare it with two computationally efficient, state-of-the-art layer-wise sparsity methods: ATP (Huang et al., 2025) and AlphaPruning (Lu et al., 2024). ATP theoretically shows that arithmetic sequences provide an effective sparsity allocation, while AlphaPruning applies Heavy-Tailed Self-Regularization Theory to assess layer importance. Since AlphaPruning published layer-wise metrics for LLaMA1 and LLaMA2, we run experiments on LLaMA2-13B. For ATP, we perform a fixed-step grid search to find the tolerance that minimizes WikiText perplexity. Results in Table 17 show that LSA outperforms both ATP and AlphaPruning on perplexity and zero-shot tasks.

## F  COMPARISON OF SPARSITY ALLOCATION BETWEEN DIFFERENT GRANULARITY

Since LSA achieves better performance with block-wise granularity than with layer-wise granularity on Llama3, we compare their sparsity allocation results at $\beta = 0.15$ in Tables 18 and 19. On

Table 17: Perplexity on different validation datasets and accuracy (%) on seven zero-shot tasks of LLaMA2-13B models at 70% unstructured sparsity. The best performance result is indicated in bold.

| Method | Layer-wise Sparsity | Perplexity↓ | | | Accuracy(%)↑ | | | | | | | |
|---|---|---|---|---|---|---|---|---|---|---|---|---|
| | | WikiText | PTB | C4 | WinoGrande | HellaSwag | BoolQ | PIQA | OBQA | ARC-e | ARC-c | Mean |
| SparseGPT | Uniform | 19.53 | 284.67 | 23.66 | 59.91 | 36.46 | 62.45 | 66.05 | 19.20 | 49.12 | 24.91 | 45.44 |
| | ATP | 13.19 | **143.76** | 15.61 | 62.75 | **42.17** | 62.87 | **70.13** | **23.40** | 51.64 | **29.69** | 48.95 |
| | ALphaPruning | 14.31 | 195.73 | 17.74 | **65.04** | 40.12 | 65.05 | 67.63 | 21.40 | 51.39 | 28.84 | 48.50 |
| | OWL | 14.31 | 196.07 | 17.48 | 64.46 | 40.04 | 64.33 | 68.55 | 21.07 | 51.01 | 27.79 | 48.18 |
| | DLP | 13.39 | 151.42 | 15.92 | 64.88 | 41.55 | 64.08 | 69.10 | 22.87 | 50.87 | 28.84 | 48.88 |
| | Ours | **12.56** | 145.60 | **15.39** | 64.85 | 41.60 | **65.62** | 68.92 | 22.87 | **52.13** | 29.41 | **49.34** |
| Wanda | Uniform | 45.26 | 445.76 | 58.45 | 52.57 | 29.28 | 62.20 | 58.38 | 12.00 | 35.48 | 18.17 | 38.30 |
| | ATP | 16.50 | 182.99 | 19.49 | 60.62 | 40.12 | **62.35** | 68.44 | **22.40** | 51.43 | **29.52** | 47.84 |
| | ALphaPruning | 17.12 | 230.28 | 21.87 | 61.09 | 38.61 | 62.23 | 67.52 | 20.93 | 51.30 | 27.73 | 46.84 |
| | OWL | 18.16 | 225.65 | 22.30 | 61.14 | 38.23 | 62.21 | 67.99 | 20.93 | 50.45 | 27.64 | 46.94 |
| | DLP | 16.80 | 153.89 | 18.87 | 61.37 | 40.41 | 62.27 | 68.41 | 21.60 | 50.70 | 28.86 | 47.66 |
| | Ours | **15.36** | **135.03** | **17.73** | **62.61** | **41.42** | 62.21 | **68.82** | 21.75 | 49.63 | 29.31 | **47.96** |

LLaMA3-8B, block allocation significantly outperformed layer allocation. We attribute this primarily to architectural differences between LLaMA1-7B, LLaMA2-7B, and LLaMA3-8B.

LLaMA3-8B incorporates the **Grouped-Query Attention (GQA) mechanism**, where multiple queries share key-value pairs. This technique was not employed in LLaMA1-7B or LLaMA2-7B. The outputs of the **k.proj and v.proj** layers are **replicated multiple times** based on the group count in GQA, increasing the effective importance of each weight in these projections. In layer-level allocation, each projection, q, k, v, uses the same sparsity rate, which is suboptimal—the sparsity rate for kv projections should be reduced. With block allocation, projections within the same block share a same score, but due to parameter count differences, the final sparsity for kv projections becomes lower than for the q projection.

In layer-grained allocation, the last layer of LLaMA1-7B and LLaMA2-7B had lower sparsity than preceding layers. However, the sparsity results allocated for LLaMA3-8B indicate that its last layer had the highest sparsity level, contradicting findings in some literature (Gromov et al., 2025b; Ma et al., 2023) suggesting **the last layer's relative importance**. Under block-wise allocation, the attention (attn) module in the last layer exhibits lower sparsity than the feed-forward network (ffn) modules in preceding layers. By **suppressing excessive pruning of the last layer**, fine-grained allocation enhances performance.

Table 18: Sparsity of LLaMA3-8B pruned with per-layer LSA at 70% unstructured sparsity, using SparseGPT (Sparsity: 70%, Perplexity: 39.56).

| Layer | q.proj | k.proj | v.proj | o.proj | gate.proj | up.proj | down.proj |
|---|---|---|---|---|---|---|---|
| 0 | 0.5520 | 0.5520 | 0.5520 | 0.5520 | 0.5520 | 0.5520 | 0.5520 |
| 1 | 0.5595 | 0.5595 | 0.5595 | 0.5595 | 0.5595 | 0.5595 | 0.5595 |
| 4 | 0.6149 | 0.6149 | 0.6149 | 0.6149 | 0.6149 | 0.6149 | 0.6149 |
| 30 | 0.8342 | 0.8342 | 0.8342 | 0.8342 | 0.8342 | 0.8342 | 0.8342 |
| 31 | 0.8520 | 0.8520 | 0.8520 | 0.8520 | 0.8520 | 0.8520 | 0.8520 |

Table 19: Sparsity of LLaMA3-8B pruned with per-block LSA at 70% unstructured sparsity, using SparseGPT (Sparsity: 70%, Perplexity:32.58).

| Layer | q.proj | k.proj | v.proj | o.proj | gate.proj | down.proj | up.proj |
|---|---|---|---|---|---|---|---|
| 0 | 0.5488 | 0.0954 | 0.0954 | 0.5488 | 0.6611 | 0.6611 | 0.6611 |
| 1 | 0.5518 | 0.1074 | 0.1074 | 0.5518 | 0.6649 | 0.6649 | 0.6649 |
| 4 | 0.5978 | 0.2913 | 0.2913 | 0.5978 | 0.6834 | 0.6834 | 0.6834 |
| 30 | 0.6475 | 0.4900 | 0.4900 | 0.6475 | 0.8074 | 0.8074 | 0.8074 |
| 31 | 0.6555 | 0.5220 | 0.5220 | 0.6555 | 0.8160 | 0.8160 | 0.8160 |

## G HYPERPARAMETER SETTING

In this section, we report the hyperparameter $\beta$ used to scale importance scores (their magnitudes vary with the sparsity ratio). To facilitate reproduction, we list the chosen values in Table 20.

Table 20: Hyperparameter settings for different sparsity levels.

| Sparsity | 10% | 20% | 30% | 40% | 50% | 60% | 70% | 80% |
|---|---|---|---|---|---|---|---|---|
| $\beta$ | 0.06 | 0.02 | 0.04 | 0.02 | 0.04 | 0.1 | 0.15 | 0.12 |

## H METRIC COMPARISON WITH WANDA

To further validate the effectiveness of our linear reconstruction error (RE) metric, we compare it against the Wanda metric. For a fair comparison, we apply uniform sparsity pruning to LLaMA models using Eq. (4) and evaluate both perplexity and zero-shot accuracy. As shown in Table 21, RE outperforms Wanda in nearly all cases. These results indicate that linear reconstruction error more reliably identifies layer importance than Wanda under uniform sparsity.

Table 21: Perplexity on different validation datasets and accuracy (%) on seven zero-shot tasks of LLaMA models at 70% unstructured sparsity. The best performance result is indicated in bold.

| Model | Method | Perplexity↓ | | | Accuracy(%)↑ | | | | | | | |
|---|---|---|---|---|---|---|---|---|---|---|---|---|
| | | WikiText | PTB | C4 | WinoGrande | HellaSwag | BoolQ | PIQA | OBQA | ARC-e | ARC-c | Mean |
| LLaMA1-7B | Wanda | 83.45 | 624.97 | 83.18 | 51.7 | 28.44 | 61.87 | 57.67 | 13.0 | 30.81 | 18.6 | 37.44 |
| | RE | **62.24** | **527.86** | **45.20** | **54.14** | **30.85** | **62.17** | **60.01** | **14.2** | **35.61** | **21.67** | **39.81** |
| LLaMA1-13B | Wanda | 53.47 | 360.95 | 55.79 | 52.49 | 30.72 | 62.17 | 61.53 | 15.4 | 37.75 | 19.2 | 39.89 |
| | RE | **34.02** | **283.67** | **23.26** | **57.77** | **37.11** | **62.42** | **68.28** | **18.6** | **50.0** | **25.34** | **45.65** |
| LLaMA1-30B | Wanda | 17.67 | 107.32 | 19.03 | **63.54** | 43.59 | **64.53** | 71.76 | **23.0** | 59.3 | 29.95 | 50.81 |
| | RE | **16.30** | **106.04** | **16.32** | 63.22 | **46.14** | 64.34 | **73.39** | 22.0 | **61.74** | **33.28** | **52.02** |
| LLaMA2-7B | Wanda | 75.01 | 308.89 | 83.53 | 49.01 | 27.84 | 59.14 | 55.82 | 11.4 | 28.79 | 18.77 | 35.82 |
| | RE | **69.26** | 403.48 | **55.64** | **52.96** | **30.02** | **62.05** | **60.34** | **15.0** | **36.78** | **20.99** | **39.73** |
| LLaMA2-13B | Wanda | **45.26** | **445.76** | 58.45 | 52.57 | 29.28 | **62.2** | 58.38 | 12.0 | 35.48 | 18.17 | 38.30 |
| | RE | 58.43 | 521.17 | **45.14** | **57.54** | **33.43** | 62.17 | **65.67** | **20.8** | **47.94** | **24.57** | **44.59** |

## I LINEAR RECONSTRUCTION ERROR PRUNING RATIO SETTING

LSA allocates sparsity to each layer by computing their minimal linear reconstruction error. The pruning ratio $p$ used in this step is a hyperparameter that directly affects the magnitude of the error. We evaluate the perplexity of LLaMA models at 70% sparsity using Wanda, where the linear reconstruction error are obtained from LSA at various pruning ratios. As shown in Table 22, perplexity values are similar for pruning ratios below 70%. Although layers' linear reconstruction errors increase with higher LSA pruning ratio $p$, each layer's error as a fraction of the total error remains stable for $p$ below 70%. These results indicate that the hyperparameter $p$ is not critical for sparsity allocation, further supporting LSA's robustness and effectiveness.

Table 22: Perplexity on the WikiText validation dataset with various linear reconstruction error pruning ratio setting using Wanda at 70% sparsity.

| Model | LSA Pruning Ratio | | | | | | | | |
|---|---|---|---|---|---|---|---|---|---|
| | 10% | 20% | 30% | 40% | 50% | 60% | 70% | 80% | 90% |
| LLaMA1-7B | 20.33 | 20.42 | 20.38 | 20.43 | 20.32 | 20.18 | 20.10 | 20.13 | 21.57 |
| LLaMA1-13B | 13.42 | 13.42 | 13.41 | 13.37 | 13.39 | 13.38 | 13.39 | 13.47 | 13.96 |
| LLaMA2-7B | 22.62 | 22.75 | 22.62 | 22.52 | 22.62 | 22.35 | 22.27 | 23.40 | 29.48 |
| LLaMA2-13B | 14.89 | 14.86 | 14.87 | 14.87 | 14.91 | 14.96 | 15.28 | 15.84 | 17.01 |
| Qwen2.5-7B | 61.04 | 60.78 | 60.59 | 60.20 | 62.08 | 64.86 | 70.95 | 77.23 | 89.29 |
| Qwen3-8B | 74.17 | 73.89 | 74.11 | 74.64 | 73.39 | 73.90 | 73.98 | 71.06 | 68.97 |

## J    COMPARED WITH LINEAR RECONSTRUCTION ERROR PRODUCED BY OTHER PRUNING METRIC

To fully validate LSA, we compare it with layer-wise sparsity allocations derived from the linear reconstruction errors produced by SparseGPT and Wanda, and evaluate perplexity on Llama1-7B pruned with SparseGPT. As shown in Table 23, LSA outperforms both SparseGPT and Wanda. These results indicate that LSA's linear reconstruction error more reliably identifies layer importance.

Table 23: Perplexity on different validation datasets and accuracy (%) on seven zero-shot tasks of LLaMA1-7B model at 70% sparsity pruned by SparseGPT. Layer-wise sparsity allocation are based on linear reconstruction error produced by SparseGPT, Wanda and LSA. The best performance result is indicated in bold.

| Layer-wise Sparsity | WikiText | PTB | C4 |
|:---:|:---:|:---:|:---:|
| +RE(Wanda) | 22.29 | 351.77 | 24.59 |
| +RE(SparseGPT) | 20.25 | 303.42 | 21.88 |
| +LSA | **17.57** | **215.61** | **19.50** |

## K    ROBUSTNESS ACROSS VARIOUS VALIDATION DATASETS

We further evaluate our method on WikiText (Merity et al., 2017), PTB (Marcus et al., 1994), and C4 (Raffel et al., 2020). We perform extensive experiments on LLaMA-1 (7B, 13B, 30B), LLaMA-2 (7B, 13B), and LLaMA-3 (8B) at 70% sparsity. As shown in Table 24, LSA combined with Wanda or SparseGPT outperforms Uniform, OWL, and DLP in nearly every case. These results demonstrate the robustness of our method across diverse validation datasets.

## L    ZERO-SHOT TASKS PERFORMANCE

For zero-shot evaluation, we measure accuracy on seven commonsense benchmarks from EleutherAI LM Harness (Gao et al., 2023), including BoolQ (Clark et al., 2019), PIQA (Bisk et al., 2019), HellaSwag (Zellers et al., 2019), WinoGrande (Sakaguchi et al., 2019), ARC-Easy (Boratko et al., 2018), ARC-Challenge (Boratko et al., 2018) and OpenbookQA (Mihaylov et al., 2018). Per-task accuracies are shown in Table 25. Overall, LSA attains the highest average accuracy in nearly every setting.

## M    LARGE LANGUAGE MODELS USAGE IN OUR PAPER

We gratefully acknowledge the use of Large Language Models linguistic editing to enhance the manuscript's read ability. The AI-generated edits were critically evaluated and modified by the author, ensuring the fidelity of the content.

Table 24: Perplexity of LLaMA models on different validation datasets at 70% unstructured sparsity. The best performance result is indicated in bold.

| Model | Method | Layer-wise Sparsity | WikiText | PTB | C4 |
|---|---|---|---|---|---|
| LLaMA1-7B | Dense | - | 5.68 | 36.42 | 7.58 |
| | Magnitude | Uniform | 48836.54 | 144955.48 | 22156.64 |
| | | OWL | 19481.62 | **110239.75** | 28196.23 |
| | | DLP | **4051.43** | 1873462.19 | 9332.01 |
| | | Ours | 4372.40 | 1232607.51 | **7608.37** |
| | SparseGPT | Uniform | 27.18 | 388.70 | 28.53 |
| | | OWL | 18.98 | 246.77 | 21.19 |
| | | DLP | 17.78 | 223.46 | 19.82 |
| | | Ours | **17.57** | **215.61** | **19.50** |
| | Wanda | Uniform | 83.45 | 624.97 | 83.18 |
| | | OWL | 24.85 | 363.84 | 27.38 |
| | | DLP | 20.89 | 276.45 | 22.90 |
| | | Ours | **20.66** | **265.05** | **22.76** |
| LLaMA1-13B | Dense | - | 5.09 | 25.27 | 7.06 |
| | Magnitude | Uniform | 84514.06 | 83469.16 | 37052.07 |
| | | OWL | 20846.71 | **98135.49** | 19514.48 |
| | | DLP | **7529.08** | 146070.26 | **32483.04** |
| | | Ours | 8682.50 | 105622.93 | 36390.94 |
| | SparseGPT | Uniform | 20.36 | 164.90 | 23.14 |
| | | OWL | 14.15 | 116.58 | 16.52 |
| | | DLP | 12.87 | 93.06 | 14.75 |
| | | Ours | **12.45** | **84.59** | **14.48** |
| | Wanda | Uniform | 53.47 | 360.95 | 55.79 |
| | | OWL | 16.49 | 162.51 | 19.40 |
| | | DLP | 13.94 | 100.94 | 16.19 |
| | | Ours | **13.65** | **91.39** | **15.94** |
| LLaMA1-30B | Dense | - | 4.10 | 21.21 | 6.39 |
| | Magnitude | Uniform | 972.99 | 7042.14 | 4807.79 |
| | | OWL | 242.56 | 1165.50 | 765.41 |
| | | DLP | 96.85 | 554.96 | 115.10 |
| | | Ours | **93.42** | **544.71** | **108.18** |
| | SparseGPT | Uniform | 12.34 | 66.30 | 15.80 |
| | | OWL | 10.69 | 49.29 | 12.92 |
| | | DLP | 9.62 | 43.16 | 11.88 |
| | | Ours | **9.49** | **41.31** | **11.80** |
| | Wanda | Uniform | 17.67 | 107.32 | 19.03 |
| | | OWL | 11.02 | 60.05 | 13.92 |
| | | DLP | 9.95 | 44.43 | 12.63 |
| | | Ours | **9.77** | **43.06** | **12.47** |
| LLaMA2-7B | Dense | - | 5.47 | 24.09 | 7.53 |
| | Magnitude | Uniform | 49799.57 | 36727.36 | 29519.90 |
| | | OWL | 59566.99 | 81089.32 | 24024.65 |
| | | DLP | **7888.52** | **38030.13** | **3646.19** |
| | | Ours | 8994.37 | 53501.48 | 3931.41 |
| | SparseGPT | Uniform | 26.32 | 839.35 | 32.58 |
| | | OWL | 20.68 | 712.58 | 23.64 |
| | | DLP | 18.68 | 212.96 | 19.97 |
| | | Ours | **18.63** | **203.36** | **19.68** |
| | Wanda | Uniform | 75.01 | 308.89 | 83.53 |
| | | OWL | 30.03 | 198.37 | 37.39 |
| | | DLP | **22.85** | 134.53 | 27.02 |
| | | Ours | 22.89 | **133.08** | **26.30** |
| LLaMA2-13B | Dense | - | 4.88 | 34.41 | 6.99 |
| | Magnitude | Uniform | 214.19 | 2472.35 | 194.01 |
| | | OWL | 59.31 | 1717.48 | 51.03 |
| | | DLP | 52.16 | 825.59 | 42.27 |
| | | Ours | **45.16** | **595.34** | **40.35** |
| | SparseGPT | Uniform | 19.53 | 284.67 | 23.66 |
| | | OWL | 14.31 | 196.07 | 17.48 |
| | | DLP | 13.39 | 151.42 | 15.92 |
| | | Ours | **12.56** | **145.60** | **15.39** |
| | Wanda | Uniform | 45.26 | 445.76 | 58.45 |
| | | OWL | 18.16 | 225.65 | 22.30 |
| | | DLP | 16.80 | 153.89 | 18.87 |
| | | Ours | **15.36** | **135.03** | **17.73** |

Table 25: Accuracy (%) of LLaMA models on seven zero-shot tasks at 70% unstructured sparsity. The best performance result is indicated in bold.

| Model | Method | Layer-wise Sparsity | WinoGrande | HellaSwag | BoolQ | PIQA | OBQA | ARC-e | ARC-c | Mean |
|---|---|---|---|---|---|---|---|---|---|---|
| LLaMA1-7B | Dense | - | 67.09 | 56.41 | 73.52 | 78.29 | 28.2 | 67.3 | 38.23 | 58.43 |
| | Magnitude | Uniform | 49.25 | 25.75 | 60.73 | 52.72 | 13.60 | 25.51 | 20.99 | 35.51 |
| | | OWL | **51.19** | 27.11 | 61.52 | 56.55 | **15.30** | 27.55 | 21.93 | 37.30 |
| | | DLP | 49.84 | 30.34 | 62.03 | 58.08 | 15.00 | 29.38 | **23.80** | 38.35 |
| | | Ours | 51.15 | **30.45** | **62.06** | **58.19** | 14.80 | **30.22** | 23.72 | **38.66** |
| | SparseGPT | Uniform | 55.56 | 34.2 | 63.18 | 64.31 | 15.6 | 42.09 | 24.32 | 42.75 |
| | | OWL | **58.60** | 36.75 | **64.77** | 64.50 | 16.50 | 42.49 | 26.02 | 44.23 |
| | | DLP | 58.46 | 37.47 | 63.91 | 64.94 | 17.13 | 42.66 | 25.23 | 44.26 |
| | | Ours | 58.49 | **38.02** | 64.60 | **65.72** | **17.27** | **43.15** | **26.14** | **44.77** |
| | Wanda | Uniform | 51.7 | 28.44 | 61.87 | 57.67 | 13.0 | 30.81 | 18.6 | 37.44 |
| | | OWL | 56.35 | 34.70 | **62.74** | 63.90 | 16.20 | 42.34 | 24.45 | 42.95 |
| | | DLP | **56.50** | 36.72 | 62.49 | **66.40** | **18.48** | **44.77** | 24.79 | **44.31** |
| | | Ours | 56.43 | **37.06** | 62.36 | 66.26 | 18.08 | 44.19 | **24.97** | 44.19 |
| LLaMA1-13B | Dense | - | 70.17 | 59.11 | 69.76 | 79.0 | 30.6 | 74.58 | 43.94 | 61.02 |
| | Magnitude | Uniform | 50.75 | 26.24 | 62.05 | 53.59 | 14.6 | 26.64 | 19.88 | 36.25 |
| | | OWL | 50.98 | **27.56** | **62.22** | **54.71** | 15.80 | 28.66 | 23.42 | 37.62 |
| | | DLP | 51.10 | 27.18 | 62.14 | 53.78 | **19.10** | 30.79 | 23.89 | 38.28 |
| | | Ours | **52.59** | 27.06 | 62.13 | 53.86 | 18.93 | **30.88** | **25.11** | **38.65** |
| | SparseGPT | Uniform | 60.77 | 36.39 | 62.63 | 66.49 | 19.8 | 47.26 | 24.91 | 45.46 |
| | | OWL | 63.53 | 40.94 | **62.77** | 69.35 | 20.47 | 52.05 | 28.02 | 48.16 |
| | | DLP | 63.71 | 42.26 | 62.28 | 70.66 | **22.00** | **52.42** | 28.17 | 48.79 |
| | | Ours | **64.59** | **42.53** | 62.19 | **70.89** | 21.68 | 51.47 | **29.03** | **48.91** |
| | Wanda | Uniform | 52.49 | 30.72 | 62.17 | 61.53 | 15.4 | 37.75 | 19.2 | 39.89 |
| | | OWL | 60.75 | 39.39 | 62.17 | 68.50 | 19.53 | 51.00 | 27.53 | 46.98 |
| | | DLP | 62.04 | 42.51 | 62.17 | 70.38 | **23.40** | 52.32 | 29.10 | 48.84 |
| | | Ours | **62.72** | **43.03** | 62.17 | 70.18 | 22.80 | **52.39** | **29.67** | **48.99** |
| LLaMA1-30B | Dense | - | 72.69 | 62.67 | 69.24 | 80.96 | 29.4 | 75.46 | 46.59 | 62.43 |
| | Magnitude | Uniform | 49.72 | 25.77 | 39.51 | 52.45 | 15.4 | 25.59 | 21.33 | 32.82 |
| | | OWL | 49.56 | 25.83 | 39.31 | 52.85 | 14.70 | 27.42 | 19.88 | 32.79 |
| | | DLP | **53.23** | 29.40 | 61.80 | 60.72 | **17.80** | 35.88 | 24.61 | 40.49 |
| | | Ours | 52.72 | **29.72** | **62.32** | **62.40** | 17.60 | **37.25** | **25.00** | **41.00** |
| | SparseGPT | Uniform | 67.72 | 44.27 | 64.71 | 72.96 | 23.4 | 61.99 | 32.76 | 52.54 |
| | | OWL | 70.17 | 46.72 | 64.58 | 73.69 | 24.90 | 61.93 | 32.72 | 53.53 |
| | | DLP | 68.94 | **48.25** | 64.68 | **73.75** | **25.20** | 60.84 | 33.88 | 53.65 |
| | | Ours | **70.48** | 48.09 | 64.29 | 73.28 | 24.73 | 61.93 | **34.61** | **53.92** |
| | Wanda | Uniform | 63.54 | 43.59 | 64.53 | 71.76 | 23.0 | 59.3 | 29.95 | 50.81 |
| | | OWL | 67.56 | 47.31 | **63.32** | 74.32 | **24.90** | **62.48** | 34.05 | 53.42 |
| | | DLP | 68.25 | 48.95 | 62.66 | 73.99 | 24.80 | 60.83 | **36.24** | **53.67** |
| | | Ours | **68.25** | **49.22** | 62.45 | 74.03 | 24.25 | 59.31 | 35.88 | 53.34 |
| LLaMA2-7B | Dense | - | 67.17 | 56.68 | 70.15 | 78.29 | 31.8 | 69.28 | 39.85 | 59.03 |
| | Magnitude | Uniform | 47.99 | 26.24 | 41.71 | 53.21 | 16.4 | 26.18 | 23.89 | 33.66 |
| | | OWL | 48.86 | 27.63 | 43.15 | 56.75 | 16.60 | 30.22 | 23.63 | 35.26 |
| | | DLP | 50.48 | 31.67 | 45.16 | 59.73 | 19.70 | 33.90 | **23.98** | 37.80 |
| | | Ours | **52.01** | **31.94** | 51.59 | **60.45** | 19.70 | **34.63** | 23.89 | **39.17** |
| | SparseGPT | Uniform | 55.96 | 33.31 | 62.32 | 62.79 | 16.4 | 38.64 | 21.16 | 41.51 |
| | | OWL | 59.41 | 36.49 | **62.55** | 64.76 | 19.00 | 43.97 | 25.23 | 44.49 |
| | | DLP | 59.96 | 38.21 | 62.26 | **67.07** | 18.20 | **44.89** | 26.28 | 45.27 |
| | | Ours | **61.01** | **38.71** | 62.27 | 66.70 | **19.20** | 44.36 | **26.48** | **45.53** |
| | Wanda | Uniform | 49.01 | 27.84 | 59.14 | 55.82 | 11.4 | 28.79 | 18.77 | 35.82 |
| | | OWL | 55.54 | 31.65 | **62.19** | 62.28 | 15.47 | 39.38 | 21.27 | 41.11 |
| | | DLP | **57.82** | 35.00 | 62.16 | 65.69 | 17.68 | 44.45 | **23.60** | 43.77 |
| | | Ours | 57.54 | **35.38** | 62.17 | 65.64 | **18.20** | **44.60** | 23.48 | **43.86** |
| LLaMA2-13B | Dense | - | 69.77 | 59.69 | 67.25 | 78.73 | 32.4 | 73.23 | 45.56 | 60.95 |
| | Magnitude | Uniform | 49.8 | 27.47 | 61.87 | 55.11 | 14.8 | 30.3 | 21.08 | 37.20 |
| | | OWL | 52.21 | 33.68 | 62.52 | 60.60 | **16.75** | 33.78 | 22.27 | 40.26 |
| | | DLP | 56.91 | 37.78 | **65.93** | **64.54** | 16.05 | 41.01 | 27.29 | 44.22 |
| | | Ours | **57.70** | **38.51** | 65.63 | 63.82 | 15.95 | **41.09** | **27.99** | **44.38** |
| | SparseGPT | Uniform | 64.09 | 40.33 | 63.39 | 68.39 | 21.2 | 51.47 | 28.16 | 48.15 |
| | | OWL | 64.46 | 40.04 | 64.33 | 68.55 | 21.07 | 51.01 | 27.79 | 48.18 |
| | | DLP | **64.88** | 41.55 | 64.08 | **69.10** | 22.87 | 50.87 | 28.84 | 48.88 |
| | | Ours | 64.85 | **41.60** | 65.62 | 68.92 | **22.87** | **52.13** | **29.41** | **49.34** |
| | Wanda | Uniform | 52.57 | 29.28 | 62.2 | 58.38 | 12.0 | 35.48 | 18.17 | 38.30 |
| | | OWL | 61.14 | 38.23 | 62.21 | 67.99 | 20.93 | 50.45 | 27.64 | 46.94 |
| | | DLP | 61.37 | 40.41 | **62.27** | 68.41 | 21.60 | **50.70** | 28.86 | 47.66 |
| | | Ours | **62.61** | **41.42** | 62.21 | 68.82 | 21.75 | 49.63 | **29.31** | 47.96 |

