# OpenReview forum: "LSA: Layer-wise Sparsity Allocation for Large Language Model Pruning Based on Minimal Linear Reconstruction Error"
_ICLR.cc/2026/Conference — ICLR 2026 Poster_

### Official Review · Reviewer_ZGQH · 2025-10-14

**Soundness:** 3
**Presentation:** 3
**Contribution:** 3
**Rating:** 4
**Confidence:** 4

**Summary:**

This paper proposes Layer-wise Sparsity Allocation (LSA), a new method for pruning large language models by measuring layer importance via minimal linear reconstruction error (LRE) rather than relying on heuristic weight scores such as those used in Wanda, OWL, or DLP. The method supports finer-grained sparsity allocation (block- and projection-level) without catastrophic performance degradation. Extensive experiments on multiple LLMs (LLaMA1/2/3, Mistral, Qwen) show that LSA achieves lower perplexity and higher zero-shot accuracy than prior methods at high sparsity (70%), while also providing measurable inference speedup.

**Strengths:**

1. The paper introduces a new way to measure layer importance directly through reconstruction error, avoiding arbitrary scoring or reduction functions.
2. LSA is conceptually simple and model-agnostic, applicable to unstructured pruning and extensible to block/projection levels.
3. Extensive experiments on multiple LLMs (LLaMA1/2/3, Mistral, Qwen) show that LSA achieves lower perplexity than prior methods at high sparsity (70%), while also providing measurable inference speedup.

**Weaknesses:**

1. While the proposed method shows clear gains in perplexity, the improvements on the seven zero-shot benchmarks appear relatively marginal compared to OWL and DLP. The authors are encouraged to provide further discussion on this part.
2. The notation in the Method section ( e.g. $S_w$, $S_x$ ) is not clearly defined, which makes the mathematical formulation somewhat difficult to follow. Additionally, the method description is overly verbose and could be streamlined for better readability and comprehension.

**Questions:**

1. The proposed LRE offers a novel perspective for measuring layer importance. However, the paper lacks a thorough comparison or discussion of how LRE relates to previously established metrics, such as Hessian-based saliency or gradient magnitude. It would strengthen the work to analyze potential correlations between LRE and these existing indicators, clarifying whether LRE captures complementary or more comprehensive information. Furthermore, while the empirical performance of LRE-based pruning is impressive, the underlying intuition and theoretical motivation behind why LRE serves as a better proxy for layer importance remain insufficiently explained.

2. It is great to see that the paper includes experiments on structured pruning and N:M sparsity. However, these results would be more convincing if they also included comparisons with existing layer-wise sparsity allocation methods such as OWL and DLP, to better contextualize the advantages of LSA in structured settings.

3. The authors claim that the sparsity ratio $p$ used to compute the linear reconstruction error is not a sensitive hyperparameter. However, the provided evidence is limited to models of similar size and within the same family. A more convincing analysis would involve evaluating across different model families and scales to demonstrate broader generalization.

---

> ### Author Response · Authors · 2025-11-18
> **Response to Reviewer ZGQH(Part I)**
>
> Dear Reviewer ZGQH:
>
> Thanks for your acknowledge of our work. In the following, we would like to address your comments point by point.
>
> > **W1.**  The improvements on the seven zero-shot benchmarks appear relatively marginal.
>
> Thank you for your valuable comment.
>
> 1) Limitations in Zero-Shot Performance Improvement
>
> The modest gains in zero-shot performance may stem from **inherent model limitations and evaluation methodology
> constraints**. During testing, Questions and options are concatenated, and selections are made by choosing the option
> with minimum perplexity.
>
> As noted in [1], pruned models may **artificially "guess correctly"** on samples where the original model was uncertain,
> leading to overestimated accuracy metrics. This phenomenon is observed in our DLP experiments: on Llama-1-7B pruned
> with SparseGPT, DLP achieved lower perplexity than OWL (17.78 < 18.98), yet showed comparable zero-shot accuracy
> (44.26 ≈ 44.23).
>
> This suggests OWL-pruned models likely benefited from fortuitous guesses on originally ambiguous samples.
>
> 2) Enhanced Performance on Advanced Models
> When evaluated on advanced models Llama3-8B, Qwen2.5-7B and Qwen3-8B,
> LSA demonstrates **significant zero-shot improvements** over both OWL and DLP, confirming its robustness across
> modern model families.
>
> Table 1. Accuracy (%) of LLaMA3-8B on seven zero-shot tasks at 70% unstructured sparsity.
>
> | LLaMA3-8B | winogrande | hellaswag | boolq     | piqa      | openbookqa | arc_easy  | arc_challenge | avg       |
> |-----------|------------|-----------|-----------|-----------|------------|-----------|---------------|-----------|
> | Dense     | 72.61      | 60.17     | 81.59     | 79.65     | 34.8       | 80.09     | 50.17         | 65.58     |
> | SparseGPT | 57.3       | 33.74     | 66.39     | 62.89     | 15.0       | 44.95     | 22.01         | 43.18     |
> | +owl      | 60.93      | 36.67     | 70.58     | 65.4      | 16.2       | 48.27     | 23.81         | 45.98     |
> | +dlp      | 61.56      | 37.54     | 70.67     | 65.13     | **19.2**   | 48.19     | **25.6**      | 46.84     |
> | +lsa      | **62.12**  | **37.93** | **74.53** | **65.89** | 18.6       | **48.95** | 25.09         | **47.59** |
> | Wanda     | 48.22      | 27.28     | 50.43     | 55.6      | 13.6       | 32.15     | 17.66         | 34.99     |
> | +owl      | 49.49      | 28.4      | **61.5**  | 57.83     | 13.4       | 35.52     | 17.66         | 37.69     |
> | +dlp      | 52.41      | 29.51     | 58.53     | 60.66     | 14.0       | 38.51     | **19.03**     | 38.95     |
> | +lsa      | **53.91**  | **30.45** | 60.49     | **60.83** | **15.0**   | **41.04** | **19.03**     | **40.11** |
>
> Table 2. Accuracy (%) of Qwen2.5-7B  on seven zero-shot tasks at 70% unstructured sparsity.
>
> | Qwen2.5-7B    | winogrande | hellaswag | boolq     | piqa      | openbookqa | arc_easy  | arc_challenge | avg       |
> |---------------|------------|-----------|-----------|-----------|------------|-----------|---------------|-----------|
> | Dense         | 73.01      | 60.04     | 85.11     | 78.78     | 33.20      | 80.47     | 47.78         | 65.48     |
> | SparseGPT     | 61.72      | 40.00     | 73.24     | 68.93     | 20.00      | 63.05     | 29.18         | 50.88     |
> | +owl          | 61.09      | 38.02     | 64.62     | 67.63     | 19.20      | 59.93     | 27.39         | 48.27     |
> | +dlp          | 61.96      | 38.31     | 67.80     | 65.40     | 18.80      | 55.98     | 26.28         | 47.79     |
> | +lsa          | 62.67      | 39.04     | **77.40** | 65.78     | 19.00      | 56.10     | 27.65         | 49.66     |
> | +lsa(block-w) | **64.33**  | **40.68** | 72.69     | **68.99** | **23.00**  | **63.34** | **29.69**     | **51.82** |
> | Wanda         | 53.04      | 30.59     | 62.02     | 61.81     | 15.80      | 45.75     | 20.56         | 41.37     |
> | +owl          | 52.17      | 30.68     | 62.17     | 62.02     | 14.60      | 45.66     | 19.37         | 40.95     |
> | +dlp          | 56.67      | 33.38     | 62.23     | 63.00     | 16.60      | 45.62     | 21.16         | 42.67     |
> | +lsa          | **56.91**  | **33.41** | **62.57** | **63.06** | **16.80**  | **46.68** | **23.29**     | **43.24** |

---

> > ### Author Response · Authors · 2025-11-18
> > **Response to Reviewer ZGQH(Part II)**
> >
> > Table 3. Accuracy (%) of Qwen3-8B  on seven zero-shot tasks at 70% unstructured sparsity.
> >
> > | Qwen3-8B           | winogrande | hellaswag | boolq     | piqa      | openbookqa | arc_easy  | arc_challenge | avg       |
> > |--------------------|------------|-----------|-----------|-----------|------------|-----------|---------------|-----------|
> > | Dense              | 67.64      | 57.12     | 86.64     | 76.88     | 31.0       | 83.54     | 55.89         | 65.53     |
> > | SparseGPT          | 62.12      | 38.57     | **73.30** | 68.72     | 21.20      | 61.45     | 29.52         | 50.70     |
> > | +owl               | 60.22      | 37.17     | 66.54     | 67.41     | 21.40      | 58.75     | 27.73         | 48.46     |
> > | +dlp               | 63.77      | 37.58     | 63.33     | 66.00     | 20.40      | 57.49     | 29.61         | 48.31     |
> > | +lsa               | **64.72**  | 38.31     | 68.44     | 67.74     | 22.40      | 59.93     | 31.31         | 50.41     |
> > | +lsa(projection-w) | 64.01      | **39.49** | 71.96     | **69.15** | **23.80**  | **62.92** | **31.48**     | **51.83** |
> > | Wanda              | 53.51      | 30.53     | 62.32     | 61.15     | 15.00      | 50.04     | 21.25         | 41.97     |
> > | +owl               | 52.09      | 29.52     | 61.99     | 61.15     | 15.20      | 47.18     | 18.86         | 40.86     |
> > | +dlp               | 55.49      | 31.90     | 62.20     | 62.02     | 16.20      | 48.48     | 23.38         | 42.81     |
> > | +lsa               | **57.54**  | **32.84** | **62.39** | **63.76** | **16.40**  | **51.60** | **24.15**     | **44.10** |
> >
> > [1] Chen et al. Streamlining Redundant Layers to Compress Large Language Models. ICLR, 2025.
> >
> > > **W2.**  The notation in the Method section (e.g. $S_W$, $S_X$) is not clearly defined. The method description is overly verbose.
> >
> > Thank you for your valuable comment. In our paper, we define $S_X=X^T X$，$S_W=W^T W$, where
> > $X$ denotes input activations and $W$ represents the weight matrix. The reconstruction error when removing channels is
> > then approximated using the submatrix sum of $S = S_X \odot  S_W$.
> >
> > For unstructured pruning, directly computing $S^{(k)} = S_X \odot (W_{k,:}^T W_{k,:})$ for all channels
> > k would incur prohibitive memory overhead. To address this isssue:
> >
> > 1) We **precompute and retain** $S_X$;
> > 2) During error matrix $e$ updates, we **compute elements on-demand**: $S^{(k)}_{i,:} = W_{k,i}(W_{k,:} \odot S_{X_{i,:}})$.
> >
> > This approach significantly reduces both memory footprint and computational load. We will enhance the final version by:
> > providing formal definitions for all variables; streamlining methodological descriptions to improve readability.
> >
> > > **Q1.**  The paper lacks a thorough comparison or discussion of how LRE relates to previously established metrics.
> > The underlying intuition and theoretical motivation behind why LRE serves as a better proxy for layer importance remain insufficiently explained.
> >
> > Thank you for your insightful comments.
> >
> > 1) Relationship of LRE to Established Metrics
> >
> > While methods like OWL and DLP employ the wanda score as their metric, this approach fundamentally measures only
> > the reconstruction error from removing individual weights. In contrast, our LRE methodology **simultaneously accounts for
> > errors induced by pruning multiple weights**, thereby provides a superior measure of layer-wise redundancy.
> >
> > Compared to Hessian-based method, LRE eliminates need for computationally expensive Hessian inversion for weight
> > updating. Compared to Gradient-magnitude method, LRE requires no backpropagation, substantially reducing memory
> > overhead.
> >
> > 2) Theoretical Motivation for LRE as Redundancy Metric
> >
> > This point has also been discussed in ATP [2], where linear layers are used to qualitatively demonstrate that
> > reconstruction error increases with deeper layers. Concurrently, prior studies [3–5] indicate that redundancy
> > escalates in deeper layers, suggesting that reconstruction error can partially reflect redundancy.
> >
> > Notably, our work diverges by:
> > - Conducting **quantitative analysis of layers' minimal reconstruction error** (LRE)
> > - Discovering that the **final layer exhibits smaller reconstruction error than preceding layers**
> >
> > This observation aligns with [3], which emphasizes that the last layer of large models is critical and should
> > not be removed.
> >
> >
> > [2] Huang et al. Determining Layer-wise Sparsity for Large Language Models Through a Theoretical Perspective.
> >
> > [3] Gromov et al. The Unreasonable Ineffectiveness of the Deeper Layers.
> >
> > [4] Sun et al. The Curse of Depth in Large Language Models.
> >
> > [5] Li et al. Mix-LN: Unleashing the Power of Deeper Layers by Combining Pre-LN and Post-LN.

---

> > > ### Author Response · Authors · 2025-11-18
> > > **Response to Reviewer ZGQH(Part III)**
> > >
> > > > **Q2.**  Experiments results on structured pruning and N:M sparsity would be more convincing if they also included comparisons with existing layer-wise sparsity allocation methods.
> > >
> > > Thank you for your valuable comment.
> > >
> > > 1) Structured Pruning Results
> > >
> > > We will provide comprehensive experimental results for structured pruning in the final version. Results at 80% sparsity
> > > are omitted since perplexity scores >1000 cease to be informative.
> > > LSA demonstrates competitive performance in structured pruning scenarios.
> > >
> > > Table 4. Perplexity of the LLaMA1-7B on WikiText dataset using LLM-Pruner.
> > >
> > > | llama-7B       | wt2        | ptb         |
> > > |----------------|------------|-------------|
> > > | LLMPruner(20%) | 11.04      | 256.94      |
> > > | +owl           | 9.96       | 139.67      |
> > > | +dlp           | 10.73      | **65.67**   |
> > > | +lsa           | **9.23**   | 161.69      |
> > > | LLMPruner(40%) | 269.36     | 1036.08     |
> > > | +owl           | **152.80** | 1418.18     |
> > > | +dlp           | 166.49     | 914.04      |
> > > | +lsa           | 157.92     | **752.84**  |
> > > | LLMPruner(60%) | 1397.30    | 3549.23     |
> > > | +owl           | 3792.64    | 4011.72     |
> > > | +dlp           | 1043.754   | 3093.68     |
> > > | +lsa           | **371.97** | **1240.52** |
> > >
> > > 2) Regarding N:M Sparsity
> > >
> > > Our implementation handles N:M constraints through: prune_n - torch.round(layer_sparsity * c).
> > >
> > > This approach yields **near-identical results across sparsity allocation methods** because:
> > >
> > > - Layer sparsity differences become statistically insignificant
> > > - The integer quantization effect
> > >
> > > Consequently, comparative evaluation with OWL/DLP under N:M sparsity provides limited discriminative value.
> > > We will include these results with proper interpretation in the final version.
> > >
> > > > **Q3.** Sparsity ratio $p$ used to compute the LRE is not a sensitive hyperparameter. The provided evidence is limited to models of similar size and within the same family.
> > >
> > > Thank you for your insightful comments. We will provide **perplexity experiments on Wikitext** for hyperparameter
> > > $p$ under Qwen2.5-7B and Qwen3-8B.
> > > When $p < 0.7$, the results **do not exhibit significant differences**.
> > > These findings will be included in our final version.
> > >
> > > Table 5. Perplexity on the WikiText validation dataset with various linear reconstruction error pruning ratio setting using Wanda at 70% sparsity.
> > >
> > > | $p$        | 0.01  | 0.1   | 0.2   | 0.3   | 0.4   | 0.5   | 0.6   | 0.7   | 0.8   | 0.9   |
> > > |------------|-------|-------|-------|-------|-------|-------|-------|-------|-------|-------|
> > > | Qwen2.5-7B | 61.69 | 61.04 | 60.78 | 60.59 | 60.20 | 62.08 | 64.86 | 70.95 | 77.23 | 89.29 |
> > > | Qwen3-8B   | 74.02 | 74.17 | 73.89 | 74.11 | 74.64 | 73.39 | 73.90 | 73.98 | 71.06 | 68.97 |

---

> > > > ### Comment · Reviewer_ZGQH · 2025-11-27
> > > >
> > > > The authors’ response has addressed most of my concerns and clarified the key points that I previously found ambiguous. Given these improvements and the additional evidence provided, I am satisfied with the revisions and will raise my score to a 6.

---

> > > > > ### Author Response · Authors · 2025-11-28
> > > > > **Response to Reviewer ZGQH**
> > > > >
> > > > > Thanks for your acknowledge of our work. Thanks again for your time and effort in reviewing our paper.

---

### Official Review · Reviewer_SSxf · 2025-10-27

**Soundness:** 2
**Presentation:** 2
**Contribution:** 2
**Rating:** 2
**Confidence:** 4

**Summary:**

This paper investigates layer-wise Sparsity Allocation (LSA) for LLM pruning, which quantifies layer-wise importance by evaluating the minimal linear reconstruction error (LSE) of each transformer layer under the assumption that 50% of its least important weights are removed.

**Strengths:**

The method supports non-uniform sparsity allocation at block- or projection-level granularity within layers, without incurring catastrophic performance degradation. It enables the assignment of distinct sparsity ratios to different projections within the same
Transformer layer, providing a finer-grained allocation that can lead to performance improvements.

**Weaknesses:**

The novelty of the proposed method may be limited. It follows traditional methods to compute the layer importance first and then allocate the sparsity for each layer according to the importance. The framework is very similar to previous works such as DLP, including the importance score and the sparsity allocation range between [pr - $\beta$, pr + $\beta$].  Some detailed part may be different, but the framework and other parts are similar to provious works. The technical contribution may be limited.

The experimental results with detailed baseline comparsion mainly focus on 70% sparsity. The results of other sparsity are very rare, and other sparsity does not have baseline results. It is better to compare with baselines under different sparsity ratios to demonstrate the general performance, not only under 70% sparsity.

It mentions to be more efficient. It is better to provide more detailed analysis about the efficiency such as complexity comparison. It is hard to see why it is more efficient, more discussions with detailed complexity demonstration can enhance this part.

In the experiemnts, the improvements seem to be marginal. The performance is very close to DLP with a very small gap. Some results are different from the original DLP paper. For example, in table 5, for sparsegpt on Llama2 7B, it reports LSA with PPL 18.63 and DLP with 18.68 PPL.  The gap is very small. And in the original DLP paper, the PPL is 18.58 under the same configuration, which is better than 18.63.  Different runs may lead to different results with small variantions. The current gap is minor, and another run may change the result leading to a better baseline. The appendix also shows that in many cases, the baselines can outperform the proposed method. The  experimental improvements  seem to be marginal, and it does not seem to be significantly better than baselines.

There are some other methods which also investigate the sparsity allocation for different layers, such as [R1,R2]. It is better to discuss the comparison with more baselines.

SparseGPT and wanda are basic llm pruning methods. There are more advanced pruning methods such as [R3], which can outperform SparseGPT with large margins under uniform sparsity. It is better to combine the proposed method with more advanced pruning methods to demonstrate the advantages.

[R1] Discovering Sparsity Allocation for Layer-wise Pruning of Large Language Models

[R2] Adaptive Layer Sparsity for Large Language Models via Activation Correlation Assessment

[R3] Fast and Effective Weight Update for Pruned Large Language Models

**Questions:**

See the weakness.

---

> ### Author Response · Authors · 2025-11-18
> **Response to Reviewer SSxf(Part I)**
>
> ##  Reviewer SSxf
>
> Dear Reviewer SSxf:
>
> Thanks for your careful review and comments! In the following, we would like to address your comments point by point.
>
> > **W1.** The novelty of the proposed method may be limited.
>
> Thank you for your insightful comments. We would like to clarify that our proposed method is not merely a direct
> extension of previous works. The key innovations of our approach are as follows:
>
> 1) Theoretical Derivation & Novel Algorithm
>
> We **theoretically derived the minimum linear reconstruction error** under
> a specified sparsity constraint and provided a concrete, feasible algorithm. This stands in contrast to OWL and DLP,
> which directly adopted existing metrics without such derivation.
>
> 2) Direct Layer Importance Assessment & Paradigm Shift
>
> We directly measure layer importance using layer-wise minimum linear reconstruction error. This approach departs
> from the original paradigm of OWL and DLP as it does **not require computing weight-wise importance scores,
> nor carefully designed reduction functions** for aggregation.
>
> By directly evaluating layer importance, our method better captures layer redundancy. This enables our approach to
> achieve comparable performance even under finer-grained sparsity allocation, whereas **OWL and DLP suffer from
> performance collapse under fine-grained allocation**. The reason lies in our method's ability to precisely characterize
> the redundancy of each projection. In contrast, OWL and DLP merely provide a trend of increasing sparsity with depth
> as the limitation of weight-wise scoring.
>
> 3) Demonstrating the Potential of Fine-Grained Allocation
>
> Our method demonstrates **the potential superiority of finer-grained sparsity allocation over layer-wise allocation**.
> This possibility was deemed infeasible by both OWL and DLP, with DLP even claiming it disrupts information flow.
>
> > **W2.** The results of other sparsity are very rare. It is better to compare with baselines under different sparsity ratios to demonstrate the general performance.
>
> Thank you for your valuable comment.
>
> 1) As shown in the table below in our paper, at **low-to-medium sparsity levels, all sparsity allocation methods exhibit
> similar** perplexity performance on WikiText.
>
> The returns on performance improvements diminish with decreasing sparsity, which aligns with common patterns observed
> in model pruning and compression. Consequently, we **focused our experiments on 70% sparsity to highlight performance
> differences** between methods.
>
> Table 1. Perplexity of LLaMA1-7B on the WikiText validation dataset with various layer-wise sparsity using Wanda at different sparsity ratios.
>
> | Method       | 10%      | 20%      | 30%      | 40%      | 50%      | 60%      | 70%       | 80%        |
> |--------------|----------|----------|----------|----------|----------|----------|-----------|------------|
> | Global       | 5.82     | 6.11     | 7.02     | 9.83     | 49.07    | 35477.43 | 28941.62  | 79951.92   |
> | Uniform      | 5.70     | 5.82     | **5.99** | 6.40     | 7.26     | 10.89    | 83.45     | 6548.71    |
> | ER           | 5.70     | 5.81     | 6.04     | 6.59     | 7.82     | 12.47    | 120.50    | 7831.16    |
> | ER-Plus      | 5.71     | 5.86     | 6.26     | 7.27     | 12.21    | 766.84   | 18190.85  | 185918.25  |
> | OWL          | 5.71     | 5.81     | 6.02     | 6.39     | 7.22     | 9.44     | 24.85     | 1024.15    |
> | AlphaPruning | **5.69** | 5.81     | 6.01     | 6.39     | 7.25     | 9.51     | 24.02     | 775.43     |
> | DLP          | 5.70     | 5.81     | 6.00     | **6.38** | **7.19** | 9.45     | 20.89     | 552.53     |
> | LSA          | 5.70     | **5.81** | 6.01     | 6.39     | 7.20     | **9.38** | **20.66** | **484.93** |
>
> 2) We further provide zero-shot performance results for Qwen2.5-7B at 60% sparsity.
>
> These results demonstrate that **LSA still achieves noticeable improvements even at moderate sparsity levels**.
> We will include these findings in the final version of our paper.

---

> > ### Author Response · Authors · 2025-11-18
> > **Response to Reviewer SSxf(Part II)**
> >
> > Table 2. Accuracy (%) of Qwen2.5-7B  on seven zero-shot tasks at 60% unstructured sparsity.
> >
> > | Qwen2.5-7B    | winogrande | hellaswag | boolq     | piqa      | openbookqa | arc_easy  | arc_challenge | avg       |
> > |---------------|------------|-----------|-----------|-----------|------------|-----------|---------------|-----------|
> > | dense         | 73.01      | 60.04     | 85.11     | 78.78     | 33.20      | 80.47     | 47.78         | 65.48     |
> > | sgpt          | 68.03      | 49.47     | 82.14     | **74.65** | 26.00      | 73.57     | 39.76         | 59.09     |
> > | +owl          | 68.51      | 48.55     | 81.77     | 74.10     | 26.80      | 73.11     | 38.14         | 58.71     |
> > | +dlp          | 68.67      | 48.24     | 81.65     | 72.63     | 26.20      | 71.76     | 37.71         | 58.12     |
> > | +lsa          | 67.25      | 48.57     | 81.96     | 73.07     | 26.40      | 72.26     | 39.93         | 58.49     |
> > | +lsa(block-w) | **69.93**  | **49.88** | **83.03** | 73.50     | **28.00**  | **73.82** | **40.53**     | **59.81** |
> > | wanda         | 66.54      | 45.67     | 79.91     | 72.03     | 25.20      | **71.84** | 36.60         | 56.83     |
> > | +owl          | 67.40      | 44.53     | 78.35     | 71.55     | 24.20      | 71.34     | 36.35         | 56.25     |
> > | +dlp          | 66.77      | 44.88     | 76.73     | 70.95     | 25.20      | 69.95     | 36.60         | 55.87     |
> > | +lsa          | 66.61      | 45.38     | 79.24     | 71.49     | 25.20      | 70.03     | 37.54         | 56.50     |
> > | +lsa(block-w) | **69.06**  | **46.35** | **81.71** | **72.52** | **25.80**  | 71.72     | **38.74**     | **57.99** |
> >
> > > **W3.** It mentions to be more efficient. It is better to provide more detailed analysis about the efficiency such as complexity comparison.
> >
> > Thank you for your valuable comment. The term "efficient" manifests in our approach in two key aspects:
> >
> > 1) Structured pruning for minimal linear reconstruction error:
> >
> > We represent the reconstruction error when pruning any channel using entry sum of submatrices of $S = (W^T W) \odot (X^T X)$.
> > This formulation **avoids substantial computational overhead** associated with
> > calculating reconstruction errors through **multiple forward passes**.
> >
> > 2) Unstructured pruning for minimal linear reconstruction error:
> >
> > Directly computing $S^{(k)} = S_X \odot (W_{k,:}^T W_{k,:})$ for all channels would consume considerable memory
> > resources and incur heavy computational burden. Instead, we compute $S^{(k)}_{i,:} = W_{k,i}(W_{k,:} \odot S_{X_{i,:}})$
> > **on the fly during the update of the error matrix** $e$, thereby reducing both memory footprint and computational load.
> >
> > > **W4.** the improvements seem to be marginal. The performance is very close to DLP with a very small gap. Some results are different from the original DLP paper.
> > Different runs may lead to different results with small variantions.
> > The appendix also shows that in many cases, the baselines can outperform the proposed method.
> >
> > Thank you for your valuable comment.
> >
> > 1) While the improvements delivered by our method may be less pronounced on older models, we **achieve substantial gains**
> > compared to DLP **on more advanced models** such as LLama3-8B, Qwen2.5-7B, and Qwen3-8B.
> >
> > Notably, on these models, OWL and DLP sometimes exhibit significant performance degradation compared to uniform sparsity.
> > This highlights limitations in reducing element-wise scores via carefully-designed aggregation functions.
> >
> > Table 3. Accuracy (%) of LLaMA3-8B on seven zero-shot tasks at 70% unstructured sparsity.
> >
> > | LLaMA3-8B | winogrande | hellaswag | boolq     | piqa      | openbookqa | arc_easy  | arc_challenge | avg       |
> > |-----------|------------|-----------|-----------|-----------|------------|-----------|---------------|-----------|
> > | Dense     | 72.61      | 60.17     | 81.59     | 79.65     | 34.8       | 80.09     | 50.17         | 65.58     |
> > | SparseGPT | 57.3       | 33.74     | 66.39     | 62.89     | 15.0       | 44.95     | 22.01         | 43.18     |
> > | +owl      | 60.93      | 36.67     | 70.58     | 65.4      | 16.2       | 48.27     | 23.81         | 45.98     |
> > | +dlp      | 61.56      | 37.54     | 70.67     | 65.13     | **19.2**   | 48.19     | **25.6**      | 46.84     |
> > | +lsa      | **62.12**  | **37.93** | **74.53** | **65.89** | 18.6       | **48.95** | 25.09         | **47.59** |
> > | Wanda     | 48.22      | 27.28     | 50.43     | 55.6      | 13.6       | 32.15     | 17.66         | 34.99     |
> > | +owl      | 49.49      | 28.4      | **61.5**  | 57.83     | 13.4       | 35.52     | 17.66         | 37.69     |
> > | +dlp      | 52.41      | 29.51     | 58.53     | 60.66     | 14.0       | 38.51     | **19.03**     | 38.95     |
> > | +lsa      | **53.91**  | **30.45** | 60.49     | **60.83** | **15.0**   | **41.04** | **19.03**     | **40.11** |

---

> > > ### Author Response · Authors · 2025-11-18
> > > **Response to Reviewer SSxf(Part III)**
> > >
> > > Table 4. Accuracy (%) of Qwen2.5-7B  on seven zero-shot tasks at 70% unstructured sparsity.
> > >
> > > | Qwen2.5-7B    | winogrande | hellaswag | boolq     | piqa      | openbookqa | arc_easy  | arc_challenge | avg       |
> > > |---------------|------------|-----------|-----------|-----------|------------|-----------|---------------|-----------|
> > > | Dense         | 73.01      | 60.04     | 85.11     | 78.78     | 33.20      | 80.47     | 47.78         | 65.48     |
> > > | SparseGPT     | 61.72      | 40.00     | 73.24     | 68.93     | 20.00      | 63.05     | 29.18         | 50.88     |
> > > | +owl          | 61.09      | 38.02     | 64.62     | 67.63     | 19.20      | 59.93     | 27.39         | 48.27     |
> > > | +dlp          | 61.96      | 38.31     | 67.80     | 65.40     | 18.80      | 55.98     | 26.28         | 47.79     |
> > > | +lsa          | 62.67      | 39.04     | **77.40** | 65.78     | 19.00      | 56.10     | 27.65         | 49.66     |
> > > | +lsa(block-w) | **64.33**  | **40.68** | 72.69     | **68.99** | **23.00**  | **63.34** | **29.69**     | **51.82** |
> > > | Wanda         | 53.04      | 30.59     | 62.02     | 61.81     | 15.80      | 45.75     | 20.56         | 41.37     |
> > > | +owl          | 52.17      | 30.68     | 62.17     | 62.02     | 14.60      | 45.66     | 19.37         | 40.95     |
> > > | +dlp          | 56.67      | 33.38     | 62.23     | 63.00     | 16.60      | 45.62     | 21.16         | 42.67     |
> > > | +lsa          | **56.91**  | **33.41** | **62.57** | **63.06** | **16.80**  | **46.68** | **23.29**     | **43.24** |
> > >
> > > Table 5. Accuracy (%) of Qwen3-8B  on seven zero-shot tasks at 70% unstructured sparsity.
> > >
> > > | Qwen3-8B           | winogrande | hellaswag | boolq     | piqa      | openbookqa | arc_easy  | arc_challenge | avg       |
> > > |--------------------|------------|-----------|-----------|-----------|------------|-----------|---------------|-----------|
> > > | Dense              | 67.64      | 57.12     | 86.64     | 76.88     | 31.0       | 83.54     | 55.89         | 65.53     |
> > > | SparseGPT          | 62.12      | 38.57     | **73.30** | 68.72     | 21.20      | 61.45     | 29.52         | 50.70     |
> > > | +owl               | 60.22      | 37.17     | 66.54     | 67.41     | 21.40      | 58.75     | 27.73         | 48.46     |
> > > | +dlp               | 63.77      | 37.58     | 63.33     | 66.00     | 20.40      | 57.49     | 29.61         | 48.31     |
> > > | +lsa               | **64.72**  | 38.31     | 68.44     | 67.74     | 22.40      | 59.93     | 31.31         | 50.41     |
> > > | +lsa(projection-w) | 64.01      | **39.49** | 71.96     | **69.15** | **23.80**  | **62.92** | **31.48**     | **51.83** |
> > > | Wanda              | 53.51      | 30.53     | 62.32     | 61.15     | 15.00      | 50.04     | 21.25         | 41.97     |
> > > | +owl               | 52.09      | 29.52     | 61.99     | 61.15     | 15.20      | 47.18     | 18.86         | 40.86     |
> > > | +dlp               | 55.49      | 31.90     | 62.20     | 62.02     | 16.20      | 48.48     | 23.38         | 42.81     |
> > > | +lsa               | **57.54**  | **32.84** | **62.39** | **63.76** | **16.40**  | **51.60** | **24.15**     | **44.10** |
> > >
> > > 2) Some results differ from those reported in the original DLP paper, primarily due to different **random seeds**.
> > >
> > > Our results in the paper represent **averages over multiple runs**, which may lead to discrepancies with the original DLP
> > > findings. Thus, while repeated runs may yield variations, our method **remains superior to DLP on average**.
> > >
> > > > **W5.** It is better to discuss the comparison with more baselines.
> > >
> > > Thank you for your valuable comment. Due to the **high time complexity of DSA[R1]**'s genetic algorithm for searching
> > > optimal aggregation functions (making it impractical for rapid results) and the **unavailability of ALS[R2]'s
> > > implementation**, we selected **two computationally efficient, state-of-the-art alternatives: ATP[1] and AlphaPruning[2]**.
> > >
> > > 1) Rationale for baseline selection:
> > >
> > > ATP theoretically demonstrates that arithmetic sequences provide effective sparsity allocation.
> > > AlphaPruning leverages Heavy-Tailed Self-Regularization Theory to analyze layer importance, with published layer-wise
> > > metrics for Llama1/2.
> > >
> > > 2) These baselines enable efficient replication and comparison. For our Llama2-13B experiments:
> > >
> > > ATP performed **fixed-step grid search** to find the tolerance minimizing WikiText perplexity.
> > > AlphaPruning used **published layer importance metrics**.
> > >
> > > Results show **LSA outperforms both SparseGPT and Wanda in perplexity and zero-shot tasks**.
> > > These findings will be included in our final version.

---

> > > > ### Author Response · Authors · 2025-11-18
> > > > **Response to Reviewer SSxf(Part IV)**
> > > >
> > > > Table 6. Perplexity of LLaMA2-13B on different validation datasets at 70% unstructured sparsity.
> > > >
> > > > | LLaMA2-13B    | wt2       | ptb        | c4        |
> > > > |---------------|-----------|------------|-----------|
> > > > | Dense         | 4.88      | 34.41      | 6.99      |
> > > > | SparseGPT     | 19.53     | 284.67     | 23.66     |
> > > > | +atp          | 13.19     | **143.76** | 15.61     |
> > > > | +alphapruning | 14.31     | 195.73     | 17.74     |
> > > > | +owl          | 14.31     | 196.07     | 17.48     |
> > > > | +dlp          | 13.39     | 151.42     | 15.92     |
> > > > | +lsa          | **12.56** | 145.60     | **15.39** |
> > > > | Wanda         | 45.26     | 445.76     | 58.45     |
> > > > | +atp          | 16.50     | 182.99     | 19.49     |
> > > > | +alphapruning | 17.12     | 230.28     | 21.87     |
> > > > | +owl          | 18.16     | 225.65     | 22.30     |
> > > > | +dlp          | 16.80     | 153.89     | 18.87     |
> > > > | +lsa          | **15.36** | **135.03** | **17.73** |
> > > >
> > > > Table 7. Accuracy (%) of LLaMA2-13B on seven zero-shot tasks at 70% unstructured sparsity.
> > > >
> > > > | LLaMA2-13B    | winogrande | hellaswag | boolq     | piqa      | openbookqa | arc_easy  | arc_challenge | avg       |
> > > > |---------------|------------|-----------|-----------|-----------|------------|-----------|---------------|-----------|
> > > > | Dense         | 69.77      | 59.69     | 67.25     | 78.73     | 32.4       | 73.23     | 45.56         | 60.95     |
> > > > | SparseGPT     | 59.91      | 36.46     | 62.45     | 66.05     | 19.2       | 49.12     | 24.91         | 45.44     |
> > > > | +atp          | 62.75      | **42.17** | 62.87     | **70.13** | **23.4**   | 51.64     | **29.69**     | 48.95     |
> > > > | +alphapruning | **65.04**  | 40.12     | 65.05     | 67.63     | 21.40      | 51.39     | 28.84         | 48.50     |
> > > > | +owl          | 64.46      | 40.04     | 64.33     | 68.55     | 21.07      | 51.01     | 27.79         | 48.18     |
> > > > | +dlp          | 64.88      | 41.55     | 64.08     | 69.10     | 22.87      | 50.87     | 28.84         | 48.88     |
> > > > | +lsa          | 64.85      | 41.60     | **65.62** | 68.92     | 22.87      | **52.13** | 29.41         | **49.34** |
> > > > | Wanda         | 52.57      | 29.28     | 62.2      | 58.38     | 12.0       | 35.48     | 18.17         | 38.30     |
> > > > | +atp          | 60.62      | 40.12     | **62.35** | 68.44     | **22.4**   | **51.43** | **29.52**     | 47.84     |
> > > > | +alphapruning | 61.09      | 38.61     | 62.23     | 67.52     | 19.40      | 51.30     | 27.73         | 46.84     |
> > > > | +owl          | 61.14      | 38.23     | 62.21     | 67.99     | 20.93      | 50.45     | 27.64         | 46.94     |
> > > > | +dlp          | 61.37      | 40.41     | 62.27     | 68.41     | 21.60      | 50.70     | 28.86         | 47.66     |
> > > > | +lsa          | **62.61**  | **41.42** | 62.21     | **68.82** | 21.75      | 49.63     | 29.31         | **47.96** |
> > > >
> > > > [R1] Discovering Sparsity Allocation for Layer-wise Pruning of Large Language Models
> > > >
> > > > [R2] Adaptive Layer Sparsity for Large Language Models via Activation Correlation Assessment
> > > >
> > > > [1] Huang et al. Determining Layer-wise Sparsity for Large Language Models Through a Theoretical Perspective.
> > > >
> > > > [2] Lu et al. AlphaPruning: Using Heavy-Tailed Self Regularization Theory for Improved Layer-wise Pruning of Large Language Models.
> > > >
> > > > > **W6.** It is better to combine the proposed method with more advanced pruning methods to demonstrate the advantages.
> > > >
> > > > Thank you for your valuable comment. We will provide experimental results incorporating the [R3] framework.
> > > >
> > > > Even when integrated with [R3], **LSA maintains superior performance over all baselines**.
> > > > These results will be incorporated into our final version.
> > > >
> > > > Table 8. Perplexity of LLaMA3-8B on different validation datasets at 70% unstructured sparsity using ADMM pruning.
> > > >
> > > > | LLaMA3-8B | wt2       | ptb       | c4        |
> > > > |-----------|-----------|-----------|-----------|
> > > > | ADMM      | 29.20     | 48.82     | 37.95     |
> > > > | +owl      | **27.36** | **41.77** | 32.34     |
> > > > | +dlp      | 28.26     | 45.77     | 31.12     |
> > > > | +lsa      | 27.97     | 45.62     | **30.09** |
> > > >
> > > > Table 9. Accuracy (%) of LLaMA3-8B on seven zero-shot tasks at 70% unstructured sparsity using ADMM pruning.
> > > >
> > > > | LLaMA3-8B | winogrande | hellaswag | boolq     | piqa      | openbookqa | arc_easy  | arc_challenge | avg       |
> > > > |-----------|------------|-----------|-----------|-----------|------------|-----------|---------------|-----------|
> > > > | ADMM      | 59.98      | 35.63     | 68.56     | 64.36     | 18.00      | 48.78     | 21.93         | 45.32     |
> > > > | +owl      | 62.83      | 38.06     | 72.81     | 65.07     | 18.20      | **52.86** | 24.91         | 47.82     |
> > > > | +dlp      | 63.30      | 38.81     | 72.84     | 66.10     | 20.00      | 51.14     | 24.49         | 48.10     |
> > > > | +lsa      | **64.80**  | **39.29** | **74.46** | **66.38** | **21.60**  | 51.05     | **25.77**     | **49.05** |
> > > >
> > > > [R3] Fast and Effective Weight Update for Pruned Large Language Models
> > > >
> > > > Finally, we hope our response has addressed your concerns. Thank you!

---

> ### Comment · Reviewer_SSxf · 2025-11-27
> **discussion**
>
> Thanks for the rebuttal. My concerns are addressed and I updated my score accordingly.

---

> > ### Author Response · Authors · 2025-11-28
> > **Response to Reviewer SSxf**
> >
> > Thanks for your acknowledge of our work. Thanks again for your time and effort in reviewing our paper.

---

### Official Review · Reviewer_GCyG · 2025-10-28

**Soundness:** 2
**Presentation:** 2
**Contribution:** 2
**Rating:** 4
**Confidence:** 4

**Summary:**

This paper proposes LSA (Layer-wise Sparsity Allocation), a novel method for pruning large language models (LLMs) that quantifies layer-wise importance by evaluating the minimal linear reconstruction error (LRE) under the assumption of removing 50% of the least important weights. LSA avoids weight scoring and empirical reduce functions, enabling non-uniform sparsity allocation at finer granularities (block- or projection-level) without performance degradation. Experiments show LSA outperforms state-of-the-art methods like OWL and DLP at high sparsity levels (e.g., 70%), achieving better performance on language modeling and zero-shot tasks while supporting efficient inference and fine-tuning.

**Strengths:**

1.LSA introduces minimal linear reconstruction error as a direct measure of layer importance, eliminating the need for weight scoring or manual reduce function design.

2.LSA consistently outperforms OWL and DLP across multiple models (e.g., LLaMA, Vicuna) and tasks (e.g., WikiText perplexity, zero-shot accuracy) at high sparsity levels (70%).

**Weaknesses:**

1.The authors claim that LSA is the first method to achieve projection-level sparsity allocation, but TRIM[1] had previously implemented even finer-grained allocation: assigning sparsity rates to rows and columns of matrices.

2.These sparsity allocation methods cannot be applied to the currently most commonly used 2:4 semi-structured pruning, and are limited to unstructured pruning only. However, unstructured pruning cannot be accelerated by GPUs, which limits the practical applications of such methods.

3.As shown in Table 6, LSA only shows slight improvements over DLP in most scenarios. Why didn't the authors conduct zero-shot task performance comparisons on Llama3? On more powerful models, the advantages of LSA over DLP might be more prominent.

[1]Beck, Florentin, William Rudman, and Carsten Eickhoff. "TRIM: Achieving Extreme Sparsity with Targeted Row-wise Iterative Metric-driven Pruning." arXiv preprint arXiv:2505.16743 (2025).

**Questions:**

As shown in Tables 1 and 2, Layer-wise allocation outperforms Block-wise in most cases on LLaMA1-7B and LLaMA2-7B. However, on LLaMA3-8B, Block-wise allocation significantly surpasses Layer-wise. Do the authors provide any insights into this discrepancy?

---

> ### Author Response · Authors · 2025-11-18
> **Response to Reviewer GCyG(Part I)**
>
> Dear Reviewer GCyG:
>
> Thank you for taking the time to read and review our paper! In the following, we would like to address your comments point by point.
>
> > **W1.** TRIM[1] had previously implemented even finer-grained allocation.
>
> Thank you for your valuable comment. Based on your suggestion, we carefully read the paper you mentioned.
>
> TRIM [1] builds on layer-wise sparsity allocation methods such as OWL and AlphaPruning [2] to further allocate sparsity
> across the rows and columns of projection matrices.
> Importantly, **the sparsity that TRIM assigns remains consistent across projections within the same Transformer layer**
> — it does not implement projection-level heterogeneous allocation — so it does not contradict our claims.
>
> **TRIM does not further allocate sparsity at block-wise or projection-wise** granularity because allocation methods
> like OWL and DLP perform poorly (they collapse) at those finer granularities.
> By contrast, our method maintains reasonable performance under projection-wise sparsity allocation and even outperforms
> layer-wise allocation on some models.
>
> Under the fine-grained sparsity scheme we implement, TRIM’s intra-projection row/column allocation method can still
> be applied as a secondary step to distribute sparsity across channels inside each projection.
>
> [1] Beck et al. TRIM: Achieving Extreme Sparsity with Targeted Row-wise Iterative Metric-driven Pruning.
>
> [2] Lu ea al. AlphaPruning: Using Heavy-Tailed Self Regularization Theory for Improved Layer-wise Pruning of Large Language Models.
>
> > **W2.** Cannot be applied to the  2:4 semi-structured pruning; limited to unstructured pruning only, unstructured pruning cannot be accelerated by GPUs, which limits the practical applications of such methods.
>
> Thank you for your valuable comment.
>
> 1) Regarding semi-structured sparsity, existing work has already focused on using mixed N:M patterns per layer to improve performance [3,4].
>
> Therefore we believe it is meaningful to apply sparsity-allocation methods to distribute mixed N:M patterns,
> and we expect future work and systems will provide support for accelerating multiple N:M formats.
>
> 2) Since the sparsity in unstructured pruning is at the weight level, we must use specialized sparse inference engines to realize inference speedups.
>
> **We use DeepSparse and nm-vllm to accelerate inference in common deployment environments (including both CPU and GPU).**
>
> For CPUs, DeepSparse supports architectures including: x86 AVX2, AVX-512, AVX-512 VNNI, and ARM v8.2+,
> which covers most Intel, AMD, and Apple M-series CPUs.
>
> For GPUs, as long as the device supports CUDA, inference acceleration can be achieved using the nm-vllm. Similarly,
> if CUDA is supported for other deployment environments, nm-vllm can also be used to achieve acceleration.
>
> [3] Sun et al. DominoSearch: Find layer-wise fine-grained N:M sparse schemes from dense neural networks.
>
> [4] Akshat et al. Accelerating LLM Inference with Flexible N:M Sparsity via A Fully Digital Compute-in-Memory Accelerator
>
> > **W3.** LSA only shows slight improvements over DLP in most scenarios. conduct zero-shot task performance comparisons on Llama3.
>
> Thank you for your insightful comments. **On the more powerful Llama3 model, the zero-shot performance of LSA has significantly improved.**
>
> Similar improvements are observed on stronger models like Qwen2.5-7B and Qwen3-8B (see the authors' general response).
> We truly appreciate your suggestions.
>
> Table 1. Accuracy (%) of LLaMA3-8B on seven zero-shot tasks at 70% unstructured sparsity.
>
> | LLaMA3-8B | winogrande | hellaswag | boolq     | piqa      | openbookqa | arc_easy  | arc_challenge | avg       |
> |-----------|------------|-----------|-----------|-----------|------------|-----------|---------------|-----------|
> | Dense     | 72.61      | 60.17     | 81.59     | 79.65     | 34.8       | 80.09     | 50.17         | 65.58     |
> | SparseGPT | 57.3       | 33.74     | 66.39     | 62.89     | 15.0       | 44.95     | 22.01         | 43.18     |
> | +owl      | 60.93      | 36.67     | 70.58     | 65.4      | 16.2       | 48.27     | 23.81         | 45.98     |
> | +dlp      | 61.56      | 37.54     | 70.67     | 65.13     | **19.2**   | 48.19     | **25.6**      | 46.84     |
> | +lsa      | **62.12**  | **37.93** | **74.53** | **65.89** | 18.6       | **48.95** | 25.09         | **47.59** |
> | Wanda     | 48.22      | 27.28     | 50.43     | 55.6      | 13.6       | 32.15     | 17.66         | 34.99     |
> | +owl      | 49.49      | 28.4      | **61.5**  | 57.83     | 13.4       | 35.52     | 17.66         | 37.69     |
> | +dlp      | 52.41      | 29.51     | 58.53     | 60.66     | 14.0       | 38.51     | **19.03**     | 38.95     |
> | +lsa      | **53.91**  | **30.45** | 60.49     | **60.83** | **15.0**   | **41.04** | **19.03**     | **40.11** |

---

> > ### Author Response · Authors · 2025-11-18
> > **Response to Reviewer GCyG(Part II)**
> >
> > > **Q1.** On LLaMA3-8B, Block-wise allocation significantly surpasses Layer-wise. Do the authors provide any insights into this discrepancy?
> >
> > Thank you for your insightful comments. We discussed this point in the paper and provided sparsity results under
> > both allocation schemes for auxiliary analysis, as shown in the table below.
> >
> > Table 2. Sparsity of LLaMA3-8B pruned with per-layer LSA at 70% unstructured sparsity, using SparseGPT (Sparsity: 70%, Perplexity: 39.56)
> >
> > | Layer | q.proj | k.proj | v.proj | o.proj | gate.proj | up.proj | down.proj |
> > |-------|--------|--------|--------|--------|-----------|---------|-----------|
> > | 0     | 0.5520 | 0.5520 | 0.5520 | 0.5520 | 0.5520    | 0.5520  | 0.5520    |
> > | 1     | 0.5595 | 0.5595 | 0.5595 | 0.5595 | 0.5595    | 0.5595  | 0.5595    |
> > | 4     | 0.6149 | 0.6149 | 0.6149 | 0.6149 | 0.6149    | 0.6149  | 0.6149    |
> > | 30    | 0.8342 | 0.8342 | 0.8342 | 0.8342 | 0.8342    | 0.8342  | 0.8342    |
> > | 31    | 0.8520 | 0.8520 | 0.8520 | 0.8520 | 0.8520    | 0.8520  | 0.8520    |
> >
> > Table 3. Sparsity of LLaMA3-8B pruned with per-block LSA at 70% unstructured sparsity, using SparseGPT (Sparsity: 70%, Perplexity:32.58).
> >
> > | Layer | q.proj | k.proj | v.proj | o.proj | gate.proj | up.proj | down.proj |
> > |-------|--------|--------|--------|--------|-----------|---------|-----------|
> > | 0     | 0.5488 | 0.0954 | 0.0954 | 0.5488 | 0.6611    | 0.6611  | 0.6611    |
> > | 1     | 0.5518 | 0.1074 | 0.1074 | 0.5518 | 0.6649    | 0.6649  | 0.6649    |
> > | 4     | 0.5978 | 0.2913 | 0.2913 | 0.5978 | 0.6834    | 0.6834  | 0.6834    |
> > | 30    | 0.6475 | 0.4900 | 0.4900 | 0.6475 | 0.8074    | 0.8074  | 0.8074    |
> > | 31    | 0.6555 | 0.5220 | 0.5220 | 0.6555 | 0.8160    | 0.8160  | 0.8160    |
> >
> > On LLaMA3-8B, block allocation significantly outperformed layer allocation.
> > We attribute this primarily to architectural differences between LLaMA1-7B, LLaMA2-7B, and LLaMA3-8B:
> >
> > 1) Introduction of GQA in LLaMA3-8B
> >
> > LLaMA3-8B incorporates the **Grouped-Query Attention (GQA) mechanism**, where multiple queries share key-value pairs.
> > This technique was not employed in LLaMA1-7B or LLaMA2-7B.
> >
> > The outputs of the **k.proj and v.proj** layers are **replicated multiple times** based on the group count in GQA,
> > increasing the effective importance of each weight in these projections. In layer-level allocation,
> > each projection (q, k, v) uses the same sparsity rate, which is suboptimal—the sparsity rate for kv projections should be reduced.
> >
> > With block allocation, projections within the same block share a same score, but due to parameter count differences,
> > the final sparsity for kv projections becomes lower than for the q projection
> >
> > 2) Suppressing Excessive Pruning of the Last Layer
> >
> > In layer-grained allocation, the last layer of LLaMA1-7B and LLaMA2-7B had lower sparsity than preceding layers.
> > However, the sparsity results allocated for LLaMA3-8B indicate that its last layer had the highest sparsity level,
> > contradicting findings in some literature [5, 6] suggesting **the last layer's relative importance**.
> >
> > Under block-wise allocation, the attention (attn) module in the last layer exhibits lower sparsity than
> > the feed-forward network (ffn) modules in preceding layers. By **suppressing excessive pruning of the last layer**,
> > fine-grained allocation enhances performance.
> >
> > While OWL and DLP also employ a block-wise allocation strategy, they show no improvement, whereas our method does.
> > This underscores the distinction between our approach and methods based on element-level scores and reduction functions.
> > Furthermore, the performance of the fine-grained allocation strategy on models employing GQA,
> > such as Qwen2.5-7B and Qwen3-8B (see the authors' general response), further supports our conjecture.
> >
> > [5] Gromov et al. The Unreasonable Ineffectiveness of the Deeper Layers.
> >
> > [6] Ma et al. LLM-Pruner: On the Structural Pruning of Large Language Models.

---

> ### Comment · Reviewer_GCyG · 2025-11-20
>
> Thank you for your insightful reply. Your analysis on why block-level outperforms layer-level in the Llama3 series is very compelling.
> However, I have a couple of follow-up questions:
> 1. According to Table 1, the performance of projection-level is lower than that of layer-level, which appears to limit its broader applicability.
>
> 2. Regarding the practical implementation of semi-structured pruning, my understanding is that NVIDIA GPUs only natively support 2:4 sparsity, and the papers you cited[1] seem to rely on specialized hardware. Furthermore, it appears that unstructured pruning is primarily accelerated on CPUs, while frameworks like nm-vllm are limited to semi-structured methods.
>
> [1]Ramachandran A, Kundu S, Raha A, et al. Accelerating llm inference with flexible n: M sparsity via a fully digital compute-in-memory accelerator[J]. arXiv preprint arXiv:2504.14365, 2025.

---

> ### Author Response · Authors · 2025-11-21
> **Response to Reviewer GCyG**
>
> Thanks for your further response.
>
> > **Q1.** According to Table 1, the performance of projection-level is lower than that of layer-level, which appears to limit its broader applicability.
>
> Thank you for the thoughtful feedback. The purpose of the Table 1 is to illustrate that, although LSA's block-level and
> projection-level granular allocations yield marginally lower performance than layer-wise allocation, they clearly
> contradict DLP's claim that finer-grained allocation disrupts information flow and leads to performance collapse.
>
> In our paper, we demonstrate that block-level granularity outperforms layer-wise allocation on Llama3-8B.
>
> Following reviewer suggestions, new experiments on Qwen2.5-7B/Qwen3-8B reveal (see the authors' general response):
> - OWL and DLP underperform significantly (sometimes worse than uniform sparsity)
> - LSA maintains robust performance
>
> For Qwen3-8B with SparseGPT pruning, projection-level allocation achieves optimal results, demonstrating its viability.
>
> Table 1. Accuracy (%) of Qwen3-8B  on seven zero-shot tasks at 70% unstructured sparsity.
>
> | Qwen3-8B           | winogrande | hellaswag | boolq     | piqa      | openbookqa | arc_easy  | arc_challenge | avg       |
> |--------------------|------------|-----------|-----------|-----------|------------|-----------|---------------|-----------|
> | Dense              | 67.64      | 57.12     | 86.64     | 76.88     | 31.0       | 83.54     | 55.89         | 65.53     |
> | SparseGPT          | 62.12      | 38.57     | **73.30** | 68.72     | 21.20      | 61.45     | 29.52         | 50.70     |
> | +owl               | 60.22      | 37.17     | 66.54     | 67.41     | 21.40      | 58.75     | 27.73         | 48.46     |
> | +dlp               | 63.77      | 37.58     | 63.33     | 66.00     | 20.40      | 57.49     | 29.61         | 48.31     |
> | +lsa               | **64.72**  | 38.31     | 68.44     | 67.74     | 22.40      | 59.93     | 31.31         | 50.41     |
> | +lsa(block-w)      | 64.25      | 39.13     | 70.98     | 68.44     | **24.00**  | 61.57     | 30.80         | 51.31     |
> | +lsa(projection-w) | 64.01      | **39.49** | 71.96     | **69.15** | 23.80      | **62.92** | **31.48**     | **51.83** |
> | Wanda              | 53.51      | 30.53     | 62.32     | 61.15     | 15.00      | 50.04     | 21.25         | 41.97     |
> | +owl               | 52.09      | 29.52     | 61.99     | 61.15     | 15.20      | 47.18     | 18.86         | 40.86     |
> | +dlp               | 55.49      | 31.90     | 62.20     | 62.02     | 16.20      | 48.48     | 23.38         | 42.81     |
> | +lsa               | **57.54**  | **32.84** | **62.39** | **63.76** | **16.40**  | **51.60** | **24.15**     | **44.10** |
>
>
> > **Q2.** NVIDIA GPUs only natively support 2:4 sparsity, and the papers you cited[1] seem to rely on specialized hardware. nm-vllm are limited to semi-structured methods.
>
> Thank you for your insightful comment.
>
> 1) N:M sparsity
>
> While current NVIDIA GPUs natively support only 2:4 sparsity patterns, emerging research[1,2] demonstrates the potential
> of mixed N:M sparsity.
>
> [1] Sun et al. DominoSearch: Find layer-wise fine-grained N:M sparse schemes from dense neural networks.
>
> [2] Akshat et al. Accelerating LLM Inference with Flexible N:M Sparsity via A Fully Digital Compute-in-Memory Accelerator
>
>
> 2) nm-vllm
>
> As documented in [Deployment Guide](https://github.com/neuralmagic/nm-vllm/blob/main/examples-neuralmagic/deploy_compressed_huggingface_models):
>
> "For unstructured sparsity, NVIDIA GPUs with compute capability ≥7.0 (V100, T4, A100) are required. For semi-structured
> sparsity or Marlin quantization, GPUs with ≥8.0 capability (Ampere+/A100+) are needed."
>
> Unstructured pruned models can leverages nm-vllm for GPU acceleration.
>
> [Implementation example](https://github.com/neuralmagic/nm-vllm/tree/main/examples-neuralmagic/sparsegpt_compress_and_deploy)
> provides production-ready acceleration for SparseGPT-pruned models (core logic implemented).
>
> The [notebook demonstration](https://github.com/neuralmagic/nm-vllm/blob/main/examples-neuralmagic/deploy_compressed_huggingface_models/Deploy_Compressed_LLMs_from_Hugging_Face_with_nm_vllm.ipynb) confirms acceleration for:
> - Unstructured pruning: [neuralmagic/phi-2-pruned50](https://huggingface.co/RedHatAI/phi-2-pruned50)
> - Semi-structured pruning:  [nm-testing/llama2.c-stories110M-pruned2.4](https://huggingface.co/RedHatAI/llama2.c-stories110M-pruned2.4)
>
> Finally, we hope our response has addressed your concerns. Thank you!

---

> > ### Comment · Reviewer_GCyG · 2025-11-27
> >
> > Thank you for the detailed response and clarification. I have raised the score to 6 to support the acceptance of this paper.

---

> > > ### Author Response · Authors · 2025-11-27
> > > **Response to Reviewer GCyG**
> > >
> > > Thanks for your acknowledge of our work. Thanks again for your time and effort in reviewing our paper.

---

### Official Review · Reviewer_tytU · 2025-10-31

**Soundness:** 3
**Presentation:** 3
**Contribution:** 3
**Rating:** 6
**Confidence:** 4

**Summary:**

This paper introduces LSA, a pruning method for large language models that measures each layer’s importance using *minimal linear reconstruction error* instead of weight-based scoring. By directly quantifying how much information each layer loses when half of its least important weights are removed, LSA assigns non-uniform sparsity ratios across layers, blocks, and even projection levels, achieving fine-grained pruning without performance collapse. Extensive experiments on LLaMA, Vicuna, Mistral, and Qwen models show that LSA consistently surpasses state-of-the-art methods like OWL and DLP in perplexity, zero-shot accuracy, and inference speed, demonstrating its robustness and generalization across architectures.

**Strengths:**

1. The proposed minimum linear reconstruction error is a meaningful and insightful contribution for both the application and analysis of pruning.

2. The proposed LSA method achieves strong performance in high-sparsity pruning on the LLaMA-1/2 model series, and the experiments are solid and convincing.

3. The paper is clearly written and easy to follow.

**Weaknesses:**

1. The LLMs used in this paper are outdated and do not reflect state-of-the-art LLMs. I think it is necessary to conduct experiments on the LLaMA‑3 family or the Qwen 2.5/3 series.

2. The manuscript omits several relevant baselines and sparsity-allocation references (e.g., EvoPress, DSA).

[1] Sieberling, O., Kuznedelev, D., Kurtic, E. & Alistarh, D. (2025). EvoPress: Accurate Dynamic Model Compression via Evolutionary Search. ICML.

[2] Li, L., Dong, P., Tang, Z., Liu, X., Wang, Q., Luo, W., Xue, W., Liu, Q., Chu, X., & Guo, Y. (2024). Discovering Sparsity Allocation for Layer-wise Pruning of Large Language Models. NeurIPS.

**Questions:**

It is necessary to conduct experiments on the LLaMA‑3 family or the Qwen 2.5/3 series.

---

> ### Author Response · Authors · 2025-11-18
> **Response to Reviewer tytU (Part I)**
>
> Dear Reviewer tytU:
>
> Thanks for your acknowledge of our work. In the following, we would like to address your comments point by point.
>
> > **W1.** The LLMs used in this paper are outdated. It is necessary to conduct experiments on the LLaMA‑3 family or the Qwen 2.5/3 series.
>
> Thank you for your valuable comment. We will provide results for Llama3-8B, Qwen2.5-7B, and Qwen3-8B at 70% sparsity.
>
> The results show that, **on these more advanced LLMs, LSA achieves a substantial improvement in zero-shot performance**.
>
> Thank you very much for the suggestion — it has helped us further improve our work.
> We will include these results in the final version.
>
> Table 1. Accuracy (%) of LLaMA3-8B on seven zero-shot tasks at 70% unstructured sparsity.
>
> | LLaMA3-8B | winogrande | hellaswag | boolq     | piqa      | openbookqa | arc_easy  | arc_challenge | avg       |
> |-----------|------------|-----------|-----------|-----------|------------|-----------|---------------|-----------|
> | Dense     | 72.61      | 60.17     | 81.59     | 79.65     | 34.8       | 80.09     | 50.17         | 65.58     |
> | SparseGPT | 57.3       | 33.74     | 66.39     | 62.89     | 15.0       | 44.95     | 22.01         | 43.18     |
> | +owl      | 60.93      | 36.67     | 70.58     | 65.4      | 16.2       | 48.27     | 23.81         | 45.98     |
> | +dlp      | 61.56      | 37.54     | 70.67     | 65.13     | **19.2**   | 48.19     | **25.6**      | 46.84     |
> | +lsa      | **62.12**  | **37.93** | **74.53** | **65.89** | 18.6       | **48.95** | 25.09         | **47.59** |
> | Wanda     | 48.22      | 27.28     | 50.43     | 55.6      | 13.6       | 32.15     | 17.66         | 34.99     |
> | +owl      | 49.49      | 28.4      | **61.5**  | 57.83     | 13.4       | 35.52     | 17.66         | 37.69     |
> | +dlp      | 52.41      | 29.51     | 58.53     | 60.66     | 14.0       | 38.51     | **19.03**     | 38.95     |
> | +lsa      | **53.91**  | **30.45** | 60.49     | **60.83** | **15.0**   | **41.04** | **19.03**     | **40.11** |
>
> Table 2. Accuracy (%) of Qwen2.5-7B  on seven zero-shot tasks at 70% unstructured sparsity.
>
> | Qwen2.5-7B    | winogrande | hellaswag | boolq     | piqa      | openbookqa | arc_easy  | arc_challenge | avg       |
> |---------------|------------|-----------|-----------|-----------|------------|-----------|---------------|-----------|
> | Dense         | 73.01      | 60.04     | 85.11     | 78.78     | 33.20      | 80.47     | 47.78         | 65.48     |
> | SparseGPT     | 61.72      | 40.00     | 73.24     | 68.93     | 20.00      | 63.05     | 29.18         | 50.88     |
> | +owl          | 61.09      | 38.02     | 64.62     | 67.63     | 19.20      | 59.93     | 27.39         | 48.27     |
> | +dlp          | 61.96      | 38.31     | 67.80     | 65.40     | 18.80      | 55.98     | 26.28         | 47.79     |
> | +lsa          | 62.67      | 39.04     | **77.40** | 65.78     | 19.00      | 56.10     | 27.65         | 49.66     |
> | +lsa(block-w) | **64.33**  | **40.68** | 72.69     | **68.99** | **23.00**  | **63.34** | **29.69**     | **51.82** |
> | Wanda         | 53.04      | 30.59     | 62.02     | 61.81     | 15.80      | 45.75     | 20.56         | 41.37     |
> | +owl          | 52.17      | 30.68     | 62.17     | 62.02     | 14.60      | 45.66     | 19.37         | 40.95     |
> | +dlp          | 56.67      | 33.38     | 62.23     | 63.00     | 16.60      | 45.62     | 21.16         | 42.67     |
> | +lsa          | **56.91**  | **33.41** | **62.57** | **63.06** | **16.80**  | **46.68** | **23.29**     | **43.24** |

---

> > ### Author Response · Authors · 2025-11-18
> > **Response to Reviewer tytU (Part II)**
> >
> > Table 3. Accuracy (%) of Qwen3-8B  on seven zero-shot tasks at 70% unstructured sparsity.
> >
> > | Qwen3-8B           | winogrande | hellaswag | boolq     | piqa      | openbookqa | arc_easy  | arc_challenge | avg       |
> > |--------------------|------------|-----------|-----------|-----------|------------|-----------|---------------|-----------|
> > | Dense              | 67.64      | 57.12     | 86.64     | 76.88     | 31.0       | 83.54     | 55.89         | 65.53     |
> > | SparseGPT          | 62.12      | 38.57     | **73.30** | 68.72     | 21.20      | 61.45     | 29.52         | 50.70     |
> > | +owl               | 60.22      | 37.17     | 66.54     | 67.41     | 21.40      | 58.75     | 27.73         | 48.46     |
> > | +dlp               | 63.77      | 37.58     | 63.33     | 66.00     | 20.40      | 57.49     | 29.61         | 48.31     |
> > | +lsa               | **64.72**  | 38.31     | 68.44     | 67.74     | 22.40      | 59.93     | 31.31         | 50.41     |
> > | +lsa(projection-w) | 64.01      | **39.49** | 71.96     | **69.15** | **23.80**  | **62.92** | **31.48**     | **51.83** |
> > | Wanda              | 53.51      | 30.53     | 62.32     | 61.15     | 15.00      | 50.04     | 21.25         | 41.97     |
> > | +owl               | 52.09      | 29.52     | 61.99     | 61.15     | 15.20      | 47.18     | 18.86         | 40.86     |
> > | +dlp               | 55.49      | 31.90     | 62.20     | 62.02     | 16.20      | 48.48     | 23.38         | 42.81     |
> > | +lsa               | **57.54**  | **32.84** | **62.39** | **63.76** | **16.40**  | **51.60** | **24.15**     | **44.10** |
> >
> > > **W2.** The manuscript omits several relevant baselines and sparsity-allocation references (e.g., EvoPress, DSA).
> >
> > Thank you for your valuable comment. Based on your suggestion, We carefully read the two papers you mentioned.
> > EvoPress [1] searches by randomly adjusting per-layer sparsity,
> > while DSA [2] uses a genetic algorithm to search for reduce functions to allocate sparsity.
> > We will add these works to our final version.
> >
> > [1] Sieberling et al. EvoPress: Accurate Dynamic Model Compression via Evolutionary Search.
> >
> > [2] Li et al. Discovering Sparsity Allocation for Layer-wise Pruning of Large Language Models.
> >
> > > **Q1.** It is necessary to conduct experiments on the LLaMA‑3 family or the Qwen 2.5/3 series.
> >
> > See the response of weakness 1.

---

> > > ### Comment · Reviewer_tytU · 2025-11-27
> > >
> > > Thanks for your detailed response. I still keep my positive score.

---

> > > > ### Author Response · Authors · 2025-11-27
> > > > **Response to Reviewer tytU**
> > > >
> > > > Thanks for your acknowledge of our work. Thanks again for your time and effort in reviewing our paper.

---

### Author Response · Authors · 2025-11-18
**General Response**

First, we thank all reviewers for their thoughtful and constructive feedback.
The time and effort you invested have significantly improved this work,
and we sincerely appreciate your valuable contributions.

While we are pleased that the contributions of our paper were positively received, the reviewers shared a common
concern: zero-shot performance is not outstanding. We provide the following general response to
address this concern comprehensively.

1) **We strengthened our analysis** by incorporating the experimental results on **Llama3-8B, Qwen2.5-7B, and Qwen3-8B**
mentioned by the reviewers. These results collectively demonstrate that for stronger models,
**LSA achieves substantial gains** over baselines such as OWL and DLP.

2) **On Qwen3-8B**, OWL and DLP suffer significant degradation in performance at uniform sparsity when using SparseGPT,
while LSA shows only a slight drop. Furthermore, **after applying a projection-wise variant of LSA,
LSA surpasses uniform sparsity**. This finding indicates that our method **captures redundancy better** than OWL and DLP,
yielding improved performance.

3) **A similar phenomenon occurs on Qwen2.5-7B**. Here, we likewise **employ a block-wise allocation scheme**
to overcome the limitations of layer-wise granularity.

Table 1. Accuracy (%) of LLaMA3-8B on seven zero-shot tasks at 70% unstructured sparsity.

| LLaMA3-8B | winogrande | hellaswag | boolq     | piqa      | openbookqa | arc_easy  | arc_challenge | avg       |
|-----------|------------|-----------|-----------|-----------|------------|-----------|---------------|-----------|
| Dense     | 72.61      | 60.17     | 81.59     | 79.65     | 34.8       | 80.09     | 50.17         | 65.58     |
| SparseGPT | 57.3       | 33.74     | 66.39     | 62.89     | 15.0       | 44.95     | 22.01         | 43.18     |
| +owl      | 60.93      | 36.67     | 70.58     | 65.4      | 16.2       | 48.27     | 23.81         | 45.98     |
| +dlp      | 61.56      | 37.54     | 70.67     | 65.13     | **19.2**   | 48.19     | **25.6**      | 46.84     |
| +lsa      | **62.12**  | **37.93** | **74.53** | **65.89** | 18.6       | **48.95** | 25.09         | **47.59** |
| Wanda     | 48.22      | 27.28     | 50.43     | 55.6      | 13.6       | 32.15     | 17.66         | 34.99     |
| +owl      | 49.49      | 28.4      | **61.5**  | 57.83     | 13.4       | 35.52     | 17.66         | 37.69     |
| +dlp      | 52.41      | 29.51     | 58.53     | 60.66     | 14.0       | 38.51     | **19.03**     | 38.95     |
| +lsa      | **53.91**  | **30.45** | 60.49     | **60.83** | **15.0**   | **41.04** | **19.03**     | **40.11** |

Table 2. Accuracy (%) of Qwen2.5-7B  on seven zero-shot tasks at 70% unstructured sparsity.

| Qwen2.5-7B    | winogrande | hellaswag | boolq     | piqa      | openbookqa | arc_easy  | arc_challenge | avg       |
|---------------|------------|-----------|-----------|-----------|------------|-----------|---------------|-----------|
| Dense         | 73.01      | 60.04     | 85.11     | 78.78     | 33.20      | 80.47     | 47.78         | 65.48     |
| SparseGPT     | 61.72      | 40.00     | 73.24     | 68.93     | 20.00      | 63.05     | 29.18         | 50.88     |
| +owl          | 61.09      | 38.02     | 64.62     | 67.63     | 19.20      | 59.93     | 27.39         | 48.27     |
| +dlp          | 61.96      | 38.31     | 67.80     | 65.40     | 18.80      | 55.98     | 26.28         | 47.79     |
| +lsa          | 62.67      | 39.04     | **77.40** | 65.78     | 19.00      | 56.10     | 27.65         | 49.66     |
| +lsa(block-w) | **64.33**  | **40.68** | 72.69     | **68.99** | **23.00**  | **63.34** | **29.69**     | **51.82** |
| Wanda         | 53.04      | 30.59     | 62.02     | 61.81     | 15.80      | 45.75     | 20.56         | 41.37     |
| +owl          | 52.17      | 30.68     | 62.17     | 62.02     | 14.60      | 45.66     | 19.37         | 40.95     |
| +dlp          | 56.67      | 33.38     | 62.23     | 63.00     | 16.60      | 45.62     | 21.16         | 42.67     |
| +lsa          | **56.91**  | **33.41** | **62.57** | **63.06** | **16.80**  | **46.68** | **23.29**     | **43.24** |

---

> ### Author Response · Authors · 2025-11-18
> **Part II**
>
> Table 3. Accuracy (%) of Qwen3-8B  on seven zero-shot tasks at 70% unstructured sparsity.
>
> | Qwen3-8B           | winogrande | hellaswag | boolq     | piqa      | openbookqa | arc_easy  | arc_challenge | avg       |
> |--------------------|------------|-----------|-----------|-----------|------------|-----------|---------------|-----------|
> | Dense              | 67.64      | 57.12     | 86.64     | 76.88     | 31.0       | 83.54     | 55.89         | 65.53     |
> | SparseGPT          | 62.12      | 38.57     | **73.30** | 68.72     | 21.20      | 61.45     | 29.52         | 50.70     |
> | +owl               | 60.22      | 37.17     | 66.54     | 67.41     | 21.40      | 58.75     | 27.73         | 48.46     |
> | +dlp               | 63.77      | 37.58     | 63.33     | 66.00     | 20.40      | 57.49     | 29.61         | 48.31     |
> | +lsa               | **64.72**  | 38.31     | 68.44     | 67.74     | 22.40      | 59.93     | 31.31         | 50.41     |
> | +lsa(projection-w) | 64.01      | **39.49** | 71.96     | **69.15** | **23.80**  | **62.92** | **31.48**     | **51.83** |
> | Wanda              | 53.51      | 30.53     | 62.32     | 61.15     | 15.00      | 50.04     | 21.25         | 41.97     |
> | +owl               | 52.09      | 29.52     | 61.99     | 61.15     | 15.20      | 47.18     | 18.86         | 40.86     |
> | +dlp               | 55.49      | 31.90     | 62.20     | 62.02     | 16.20      | 48.48     | 23.38         | 42.81     |
> | +lsa               | **57.54**  | **32.84** | **62.39** | **63.76** | **16.40**  | **51.60** | **24.15**     | **44.10** |

---

### Author Response · Authors · 2025-11-29
**Submission of Revised Manuscript**

We sincerely thank the Area Chair and all reviewers for their time, detailed feedback, and positive reassessment of our work. We are encouraged by the reviewers’ acknowledgment of our rebuttal and the subsequent score increases (**[6,4,4,2] -> [6,6,6,4]**).

In response to the constructive suggestions, we have uploaded a revised version of the manuscript with significant updates **marked in red**. The key improvements are summarized below:

1) **Extensive Evaluation on Advanced Models**: Addressing the common concern regarding zero-shot performance, we have incorporated comprehensive experiments on **Llama3-8B, Qwen2.5-7B, and Qwen3-8B**. These results demonstrate that LSA achieves substantial gains over strong baselines (OWL and DLP) on these modern architectures, proving the method's robustness.

2) **Validation of Fine-Grained Allocation**: Our new analysis on Qwen3-8B reveals that while OWL and DLP suffer significant performance degradation compared to uniform sparsity, LSA maintains stability. Furthermore, by applying a projection-wise variant, LSA surpasses uniform sparsity, confirming its superior ability to capture redundancy at a finer granular level. We observed similar trends on Qwen2.5-7B, where we successfully employed a block-wise allocation scheme to **overcome the limitations of layer-wise granularity**, further validating our approach.

3) **Expanded Baselines and Comparisons**: As requested, we have added discussions and comparisons with additional baselines (e.g., ATP, AlphaPruning) and integrated our method with advanced pruning frameworks (e.g., ADMM) to demonstrate broad applicability.

We believe these revisions have significantly strengthened the paper. We remain available to answer any further questions during the final discussion phase.

---

### Meta-Review · Area_Chair_btqv · 2026-01-03

**Summary:**

This paper introduces LSA, a layer-wise sparsity allocation method for weight pruning of large language models. The key idea is to determine per-layer sparsity by quantifying layer-wise importance through the minimal linear reconstruction error (LSE) of each layer, enabling non-uniform sparsity allocation at either block-wise or projection-level granularity.

All reviewers agree that the proposed LSA method represents a meaningful contribution relative to existing approaches. Before the rebuttal, reviewers raised concerns regarding zero-shot performance, fine-grained sparsity allocation, evaluation on more recent model families, coverage of additional baselines, application to structural pruning methods, and the depth of analysis. During the rebuttal, the authors made substantial efforts to address these concerns by providing additional results, including new comparison experiments on Qwen-2.5-7B, LLaMA-3-8B, and Qwen-3-8B. Following the rebuttal, reviewers generally agree that most of their concerns have been addressed and indicate a tendency to increase their scores.

After carefully reviewing the paper, the reviews, and the rebuttal, the Area Chair agrees with the reviewers that LSA represents a meaningful contribution to the problem of sparsity allocation and therefore recommends acceptance of the paper as a poster presentation.

**Reviewer Concerns:**

Before the rebuttal, reviewers raised concerns regarding zero-shot performance, fine-grained sparsity allocation, evaluation on more recent model families, coverage of additional baselines, application to structural pruning methods, and the depth of analysis.

After the rebuttal, reviewers believe most of the concerns have been addressed based on the newly presented results from the rebuttal.

**Reviewer Scores:**

Reviewer tytU will keep their score of 6.
Reviewer GCyG will increase their score from 4 to 6 as mentioned in their comment.
Reviewer SSxf will increase their score from 2 to 4 as mentioned in their comment.
Reviewer ZGQH will increase their score from 4 to 6 as mentioned in their comment.

---

### Decision · Program_Chairs · 2026-01-26

Accept (Poster)